# Defective AMH signaling disrupts GnRH neuron development and function and contributes to hypogonadotropic hypogonadism

Samuel Andrew Malone[1,2], Georgios E Papadakis[3], Andrea Messina[3], Nour El Houda Mimouni[1,2], Sara Trova[1,2], Monica Imbernon[1,2], Cecile Allet[1,2], Irene Cimino[1], James Acierno[3], Daniele Cassatella[3], Cheng Xu[3], Richard Quinton[4], Gabor Szinnai[5], Pascal Pigny[6], Lur Alonso-Cotchico[7], Laura Masgrau[7,8], Jean-Didier Maréchal[7], Vincent Prevot[1,2], Nelly Pitteloud[3]*, Paolo Giacobini[1,2]*

[1]Jean-Pierre Aubert Research Center (JPArc), Laboratory of Development and Plasticity of the Neuroendocrine Brain, Inserm, UMR-S 1172, Lille, France; [2]University of Lille, FHU 1, 000 Days for Health, Lille, France; [3]Faculty of Biology and Medicine, Service of Endocrinology, Diabetology and Metabolism, University Hospital, Lausanne, Switzerland; [4]Institute of Genetic Medicine, University of Newcastle-upon-Tyne, Newcastle-upon-Tyne, United Kingdom; [5]Pediatric Endocrinology and Diabetology, University of Basel Children's Hospital, Basel, Switzerland; [6]CHU Lille, Laboratoire de Biochimie et Hormonologie, Centre de Biologie Pathologie, Lille, France; [7]Departament de Química, Universitat Autònoma de Barcelona, Bellaterra, Spain; [8]Institut de Biotecnologia i de Biomedicina, Universitat Autònoma de Barcelona, Bellaterra, Spain

*For correspondence:
Nelly.Pitteloud@chuv.ch (NP);
paolo.giacobini@inserm.fr (PG)

Competing interests: The authors declare that no competing interests exist.

**Abstract** Congenital hypogonadotropic hypogonadism (CHH) is a condition characterized by absent puberty and infertility due to gonadotropin releasing hormone (GnRH) deficiency, which is often associated with anosmia (Kallmann syndrome, KS). We identified loss-of-function heterozygous mutations in anti-Müllerian hormone (*AMH*) and its receptor, *AMHR2*, in 3% of CHH probands using whole-exome sequencing. We showed that during embryonic development, AMH is expressed in migratory GnRH neurons in both mouse and human fetuses and unconvered a novel function of AMH as a pro-motility factor for GnRH neurons. Pathohistological analysis of *Amhr2-*deficient mice showed abnormal development of the peripheral olfactory system and defective embryonic migration of the neuroendocrine GnRH cells to the basal forebrain, which results in reduced fertility in adults. Our findings highlight a novel role for AMH in the development and function of GnRH neurons and indicate that AMH signaling insufficiency contributes to the pathogenesis of CHH in humans.
DOI: https://doi.org/10.7554/eLife.47198.001

## Introduction

Gonadotropin releasing hormone (GnRH) is essential for puberty onset and reproduction. GnRH is released into the pituitary portal blood vessels for delivery to the anterior pituitary. There, GnRH controls the production and release of the gonadotropins LH (luteinizing hormone) and FSH (follicle stimulating hormone), which in turn stimulate gametogenesis and sex steroid production in the gonads (*Christian and Moenter, 2010*). GnRH–secreting neurons are unusual neuroendocrine cells,

as they originate in the nasal placode outside the central nervous system during embryonic development, and migrate to the hypothalamus along the vomeronasal and terminal nerves (VNN, TN) (*Wray et al., 1989*; *Schwanzel-Fukuda and Pfaff, 1989*). This process is evolutionarily conserved and follows a similar spatio-temporal pattern in all mammals (*Wray et al., 1989*; *Schwanzel-Fukuda and Pfaff, 1989*), including humans (*Schwanzel-Fukuda et al., 1996*; *Casoni et al., 2016*). Disruption of GnRH neuronal migration and/or defective GnRH synthesis and secretion leads to congenital hypogonadotropic hypogonadisms (CHH), a rare endocrine disorder (prevalence: 1 in 4000) characterized by absent or incomplete puberty resulting in infertility (*Boehm et al., 2015*). CHH is clinically and genetically heterogeneous with several causal genes identified to date (*Boehm et al., 2015*), and follows various modes of transmission, including oligogenic inheritance (*Sykiotis et al., 2010*). However, the mutations identified so far only account for half of clinically reported cases, suggesting that other causal genes remain to be discovered. Unravelling new genetic pathways involved in the regulation of the development of the GnRH system is relevant for understanding the basis of pathogenesis leading to CHH in humans.

AMH is a TGF-β family member and it signals by binding to a specific type II receptor (AMHR2) (*di Clemente et al., 1994*; *Baarends et al., 1994*), which heterodimerizes with one of several type I TGF-β receptors (Acvr1 [Alk2], Bmpr1a [Alk3] and Bmpr1b [Alk6]), to recruit Smad proteins that subsequently undergo nuclear translocation to regulate target gene expression (*Josso and Clemente, 2003*). Although AMH signaling has been traditionally reported to play a crucial role during sex differentiation and gonadal functions (*Josso et al., 1998*; *Behringer et al., 1994*), accumulating evidence has started to shed light on unexpected functions of AMH in the central nervous system as well as in the pituitary (*Lebeurrier et al., 2008*; *Wang et al., 2009*; *Tata et al., 2018*; *Cimino et al., 2016*; *Garrel et al., 2016*). We have previously shown that GnRH neurons express AMHR2 from early fetal development to adulthood and that AMH stimulates GnRH neuronal activity and hormone secretion in mature GnRH cells (*Cimino et al., 2016*). Here, we expand this information by demonstrating that GnRH cells also express AMH during their migratory process, both in mice and human fetuses and we describe a novel role of AMH as a potent stimulator of GnRH cell motility. Finally, we show that pharmacological or genetic invalidation of Amhr2 signaling in vivo alters GnRH migration and the projections of VNN/TN to the basal forebrain, which results in a reduced size of this neuronal population in adult brains, altered ovulation and fertility. The involvement of the AMH signaling pathway in GnRH ontogeny and secretion led to the identification of four heterozygous loss-of-function mutations in *AMH* and *AMHR2* among 136 CHH patients. Collectively, this study identified a novel embryonic role of AMH in the development and function of GnRH neurons and provides genetic evidence that disturbance of AMH signaling can contribute to CHH phenotype in humans.

## Results

### Amh is expressed by GnRH migratory cells in mouse and human fetuses

We have recently shown that migratory GnRH neurons and developing vomeronasal/olfactory axons express Amhr2 in mammals (*Cimino et al., 2016*). In this study, we investigated whether migratory GnRH neurons expressed Amh in addition to Amhr2. In order to do so, we first isolated GnRH neurons through fluorescence activated cell sorting (FACS) from *Gnrh1 <GFP>* embryos (*Spergel et al., 1999*) at embryonic day 12.5 (E12.5), coincident with the beginning of the GnRH neuronal migratory process (*Wray et al., 1989*; *Schwanzel-Fukuda and Pfaff, 1989*), at postnatal day 12 (PN12) and at postnatal day 90 (PN90; *Figure 1a,b*). These experiments revealed expression of *Amh* already at E12.5 both in GnRH neurons and in total head extracts (*Figure 1a*). Moreover, real-time PCR experiments showed increasing expression of *Amh* in GnRH neurons from early embryonic development (E12.5) to adult life (PN90; *Figure 1a,b*).

Ex vivo cultures of embryonic nasal explants have been used to study factors regulating GnRH migration by both our group (*Giacobini et al., 2004*; *Giacobini et al., 2007*) and others (*Fueshko and Wray, 1994*). At 4 days in vitro, olfactory axons, which express βIII−tubulin (TUJ1), emerge from the nasal explant tissue mass and GnRH neurons begin migrating out from the explant, tightly associated to those fibers (*Figure 1c*). We generated nasal explants from *Gnrh1 <GFP>* embryos and we immunostained these primary cultures using an antibody directed against the biologically active form of Amh (C-terminal region; *Figure 1d–f*). These experiments revealed the

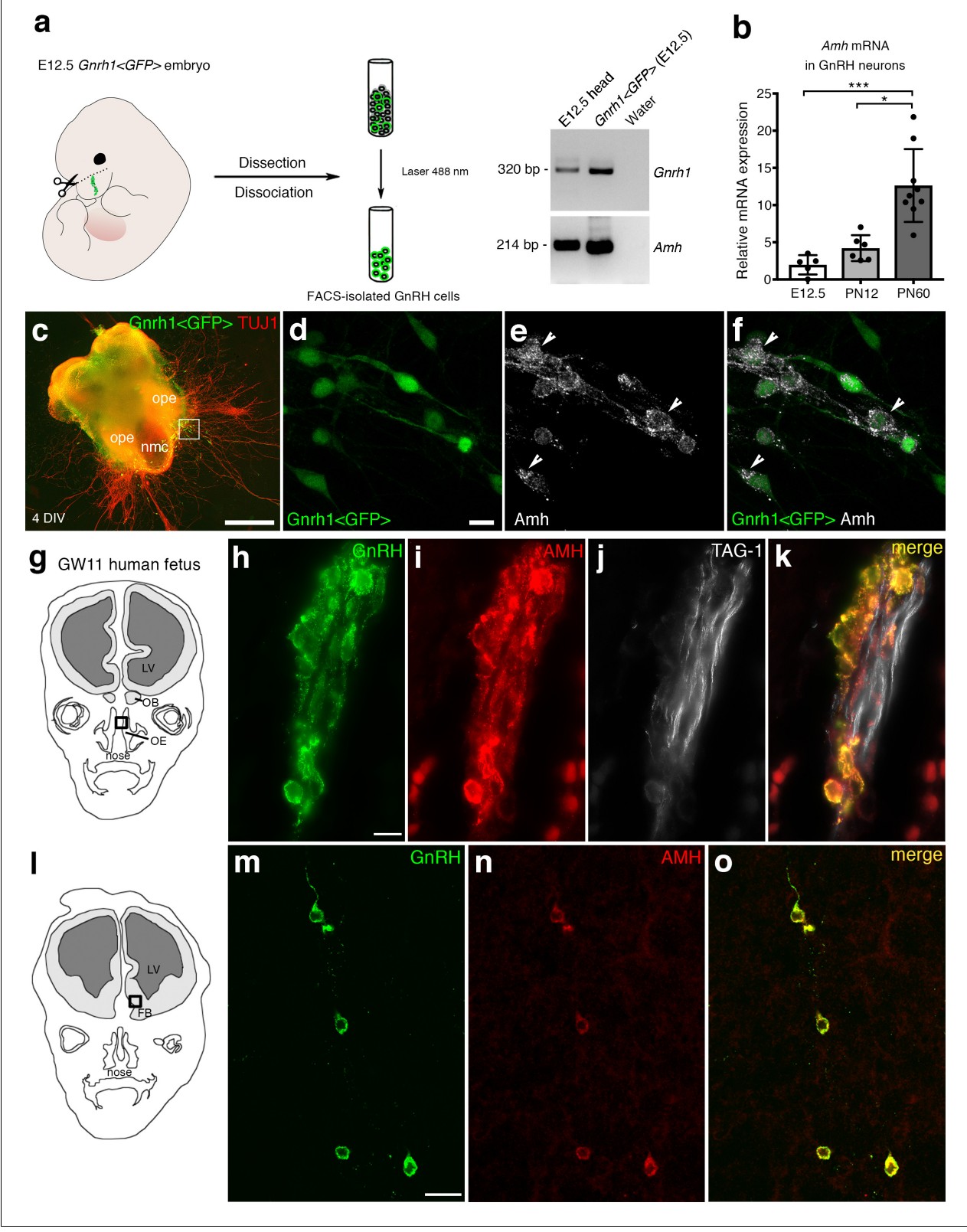

**Figure 1.** AMH is expressed in migratory GnRH neurons in mouse and human fetuses. (**a**) Schematic illustrates isolation of *Gnrh1 <GFP>* expressing cells in the nasal region of embryonic day 12.5 animals (E12.5) through fluorescent activated cell sorting (FACS). Gel on the right-hand side is a representative qualitative PCR depicting *GnRH* and *Amh* expression in migratory GnRH cells and in the head of E12.5 *Gnrh1 <GFP>* embryos. (**b**) Quantitative analysis of *Amh* mRNA expression in FACS-isolated GnRH neurons at E12.5 (*n* = 5), postnatal day 12 (PN12, *n* = 6) and postnatal day 60

*Figure 1 continued*

(PN60, *n* = 9). Data are represented as median values with the 25th-75th percentile range. Comparisons between groups were performed using a Kruskal-Wallis test followed by Dunn's post hoc analysis. *p = 0.0398, ***p = 0.0006. (**c**) Representative image of a nasal explant (out of *n* = 3) generated from a *Gnrh1 <GFP>* embryo and cultured for 4 days (DIV: days in vitro) before immunostaining for tubulin βIII (TUJ1, red). (**d–f**) Higher magnification picture of inset in **d**) showing migratory GFP-positive GnRH neurons (green) expressing Amh (white). (**g**) Schematic representation of a GW11 human fetus head (coronal view) illustrating the nasal area (box) used for immunofluorescence. (**h–k**) GnRH (green), AMH (red) and TAG-1 (white) expression in a coronal section of a GW11 fetus (out of *n* = 2 GW11 fetuses, females). AMH is expressed in GnRH neurons but not on vomeronasal/terminal fibers. (**l**) Schematic representation of a GW11 human fetus head (coronal view) illustrating the forebrain area (box) used for immunofluorescence. (**m–o**) AMH is expressed in GnRH neurons that are migrating in the forebrain. NMC: nasal midline cartilage; OPE: olfactory placode epithelium; VNO: vomeronasal organ; OE: olfactory epithelium; OB: olfactory bulb; FB: forebrain; LV: lateral ventricle. Scale bars: (**c**) 500 μm; (**d–f**) 10 μm; (**h–k**) 10 μm; (**m–o**) 20 μm.

DOI: https://doi.org/10.7554/eLife.47198.002

The following source data is available for figure 1:

**Source data 1.** This spreadsheet contains the normalized values used to generate the bar plots shown in *Figure 1b*.
DOI: https://doi.org/10.7554/eLife.47198.003

presence of Amh-immunoreactivity in punctated structures resembling vesicles (arrowheads in *Figure 1e,f*) in migratory neurons.

In order to determine whether this expression pattern was evolutionarily conserved, we next evaluated the expression of AMH in GnRH neurons and along their migratory route during human fetal development at 11th gestational week (GW11) (see schematics in *Figure 1g,l*). Triple-immunofluorescence staining of coronal sections of GW11 fetuses (*n* = 2 females) revealed that AMH is expressed in GnRH neurons but not on the vomeronasal/terminal nerves (TAG-1-positive) that form the migratory scaffold for GnRH neurons (*Figure 1h–k*). We further evaluated whether AMH expression was retained by all GnRH neurons that entered the brain (*Figure 1m–o*). Interestingly, at this developmental stage the only neurons expressing AMH in the forebrain were the GnRH neuroendocrine cells (*Figure 1m–o*). These data show that GnRH neurons start expressing Amh during their migratory process and maintain this expression until adulthood.

## Pharmacological and genetic invalidation of Amhr2 disrupts GnRH neuronal migration and the olfactory axonal scaffold

Given the expression pattern of Amh and Amhr2 along the GnRH migratory pathway (*Cimino et al., 2016*), we next investigated whether Amh could play a role on the development of the GnRH and olfactory/vomeronasal system. As the expression of Amhr2 is a prerequisite for tissues to be responsive to the actions of Amh, we investigated whether acute pharmacological blockade of the receptor with an Amhr2 neutralizing antibody (Amhr2-NA) affects the development of the olfactory, vomeronasal and terminal systems and the GnRH migration. This was achieved by in utero injection of Amhr2-NA delivered into the olfactory pit of E12.5 embryos at the beginning of the migratory process, and subsequent analysis of GnRH migration and its axonal scaffold 48 hr later (*Figure 2a*). Correct injection site in the olfactory pits was validated using the Fluorogold tracer (*Figure 2b*).

We analyzed the number and distribution of GnRH neurons in E14.5 embryos, when the GnRH population is equally distributed in the nose and in the forebrain (*Wray et al., 1989*; *Schwanzel-Fukuda and Pfaff, 1989*). At this stage, in control embryos GnRH neurons were located in the nose at the levels of the nasal/forebrain junction (N/FB J) and in the ventral forebrain (vFB; *Figure 2c,e*). Notably, while GnRH cells normally turn ventrally toward the basal forebrain in control embryos (*Figure 2c,e*), in Amhr2-NA embryos fewer neurons reached the vFB region (*Figure 2d,f*) and several GnRH cells were found scattered in ectopic cortical regions (*Figure 2d*, arrows). At E14.5, the total number of GnRH neurons was comparable between control and Amhr2-NA-treated embryos (*Figure 2g*), indicating that Amhr2 neutralization had no effect on GnRH neuron survival. However, a significant accumulation of GnRH cells in the nasal compartment, concomitant to decreased cell numbers within the vFB, is suggestive of a delayed GnRH cell migration in Amhr2-NA injected embryos (*Figure 2h*).

Immunolabeling with peripherin, a neuron-specific intermediate filament protein expressed by rodent sensory and autonomic axons (*Parysek and Goldman, 1988*), including the developing olfactory nerve (ON) and VNN (*Casoni et al., 2016*; *Fueshko and Wray, 1994*), was used to assess the

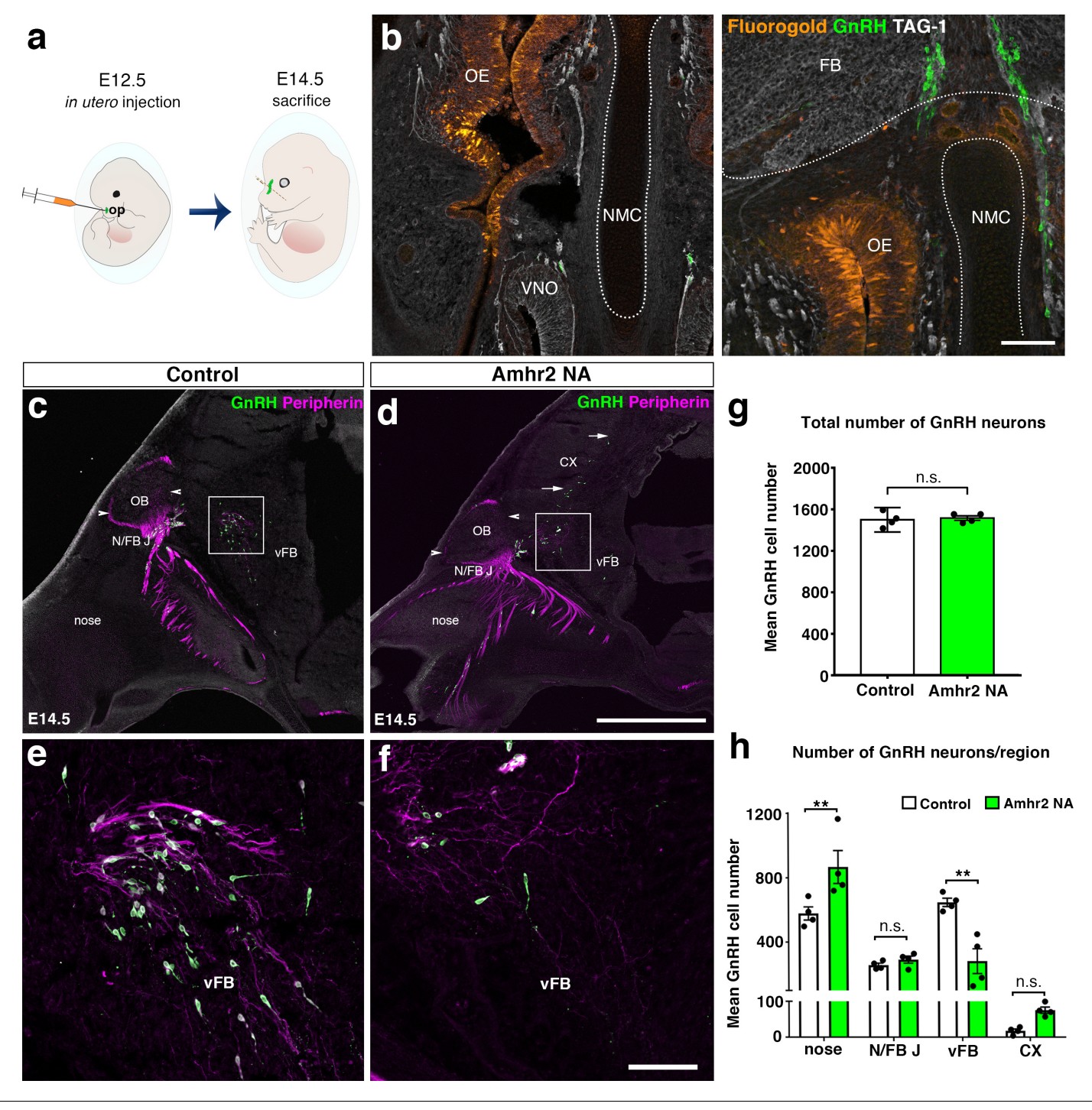

**Figure 2.** In utero pharmacological invalidation of Amhr2 disrupts GnRH neuronal migration and the olfactory/terminal nerve targeting. (a) Schematic of in utero injections targeting the olfactory pits. Injections were performed at E12.5 and embryos harvested 48 hr later. (b) Representative coronal section of an embryo head at E14.5 showing that olfactory pit Fluorogold delivery at E12.5 was successful. GnRH immunoreactive neurons are shown in green. (c–f) Representative photomicrographs of sagittal sections of mouse embryos injected at E12.5 with either saline or a neutralizing antibody for Amhr2 (Amhr2-NA) and immunostained for GnRH (green) and Peripherin (magenta) at E14.5. (e, f) Higher magnification confocal photomicrograph of boxed areas in c and d. (g) Quantification of the total number of GnRH immunoreactive neurons in saline-injected (control) and Amhr2-NA injected embryos ($n = 4$ for both groups, harvested from two independent dams). Data are represented as mean ± s.e.m ($n = 4$, unpaired two-tailed Student's $t$ test: mean cell number, $t_6 = 0.3796$, $p = 0.7173$). (h) Quantitative analysis of GnRH neuronal distribution throughout the migratory pathway in the two experimental groups. Data are represented as mean ± s.e.m ($n = 4$, two-way ANOVA, $F_{3,24} = 15.09$, p<0.0001; followed by Holm-Šídák multiple

*Figure 2 continued on next page*

*Figure 2 continued*

comparison *post hoc* test, **p<0.005; n.s., not significant; N/FB J *Amhr2*$^{+/+}$ vs. N/FB J *Amhr2*$^{-/-}$p = 0.99, CX *Amhr2*$^{+/+}$ vs. CX *Amhr2*$^{-/-}$p = 0.88. Cx: cortex; FB: forebrain; N/FBJ: nasal/forebrain junction; oe: olfactory epithelium; NMC: nasal mesenchyme. Scale bars: (**b**) 100 µm; (**d**) 2.5 mm; (**f**) 50 µm.

DOI: https://doi.org/10.7554/eLife.47198.004

The following source data is available for figure 2:

**Source data 1.** This spreadsheet contains the values used to generate the bar plots shown in *Figure 2g and h*.

DOI: https://doi.org/10.7554/eLife.47198.005

development of the ON and VNN at E14.5 (*Figure 2c–f*). ON/VNN development progressed as previously reported (*Yoshida et al., 1995*) in saline injected groups (Control); however, abnormal ON/VNN targeting occurred in embryos injected with Amhr2-NA. In these embryos, the axonal innervation of the olfactory bulb (OB) appeared incomplete as compared to controls (*Figure 2c,d*, arrowheads). This difference in axonal targeting was especially evident for the intracranial branch of the VNN projecting to the ventral forebrain (vFB; boxes in *Figure 2c,d*). In control animals, normal targeting of peripherin-positive fibers was seen as they turn ventrally to target the hypothalamus (*Figure 2e*), whereas in Amhr2-NA injected embryos the fibers had a scattered appearance (*Figure 2f*) and failed to penetrate properly into the vFB.

In light of these results, we sought to determine whether genetic invalidation of *Amhr2* would lead to similar defects. We performed a detailed analysis of E13.5 wild type and *Amhr2*$^{-/-}$ embryos using whole mount immunostaining for GnRH and peripherin followed by iDISCO tissue-clearing (*Renier et al., 2014*) and light sheet microscopy (LSM) (*Figure 3a*; *Figure 3—video 1*). In *Amhr2*$^{+/+}$ embryos, peripherin-positive fibers were seen to innervate almost completely the OB (*Figure 3b, f, h and j*, arrowheads), whereas in *Amhr2*$^{-/-}$ mice olfactory axons only partially innervated their target tissues (*Figure 3c, g, i and k*, arrowheads). Moreover, whereas in *Amhr2*$^{+/+}$ embryos GnRH neurons entered the brain along the TN projections and migrated to the ventral forebrain (vFB; *Figure 3d and j*, arrows), in *Amhr2*$^{-/-}$ embryos GnRH neurons appeared more clustered in the nasal compartment, stuck in proximity to the VNO, and fewer GnRH cells reached the vFB at this embryonic stage (*Figure 3e and k*, arrows).

Altogether, these experiments revealed that pharmacological or genetic invalidation of Amhr2 leads to abnormal development of the olfactory system, aberrant intracranial projections of the vomeronasal nerve (terminal nerve) and defective GnRH migration to the basal forebrain.

## Adult *Amhr2*-deficient mice show decreased GnRH cell number, LH secretion and fertility

To determine whether the delayed GnRH migratory process observed in *Amhr2* deficient embryos would result in a reduced number of GnRH neurons in adulthood, we immunostained for GnRH brains harvested from adult *Amhr2*$^{+/+}$ and *Amhr2*$^{-/-}$ animals. Knock-out mice had decreased GnRH immunoreactivity at the level of the organum vasculosum laminae terminalis (OVLT; *Figure 4a–d*, arrows), where the majority of GnRH cell bodies are located, as well as in the median eminence (ME) of the hypothalamus (*Figure 4e–h*), which is the projection site of neuroendocrine GnRH cells. When we counted the total number of GnRH-positive cells in *Amhr2*$^{+/+}$, *Amhr2*$^{+/-}$ and *Amhr2*$^{-/-}$ female brains, we found no difference between wild type and heterozygous mice, while we observed a significant 40% reduction in GnRH cell number in *Amhr2*$^{-/-}$ mice as compared to the other genotypes (*Figure 4i*). Male and female homozygous animals showed a similar GnRH cell loss as compared to sex-matched wild-type littermates (*Figure 4—figure supplement 1*).

Since LH secretion is an indirect measurement of GnRH neuronal secretion, we measured LH in adult female mice. Circulating LH was found to be significantly lower in *Amhr2*$^{+/-}$ and *Amhr2*$^{-/-}$ animals as compared to wild-type littermates (*Figure 4j*), supporting an impairment of GnRH secretion in these animals. However, only *Amhr2*$^{-/-}$ mice exhibited reduced ovulation, as shown by the presence of fewer post-ovulation corpora lutea in the ovaries of *Amhr2*$^{-/-}$ mice as compared to *Amhr2*$^{+/-}$ and wild-type animals (*Figure 4k*).

We then evaluated LH pulsatility by serial blood sampling in female diestrous mice (*Figure 4l,m*) and found that both *Amhr2*$^{+/-}$ and *Amhr2*$^{-/-}$ animals had a significantly lower LH pulse frequency as

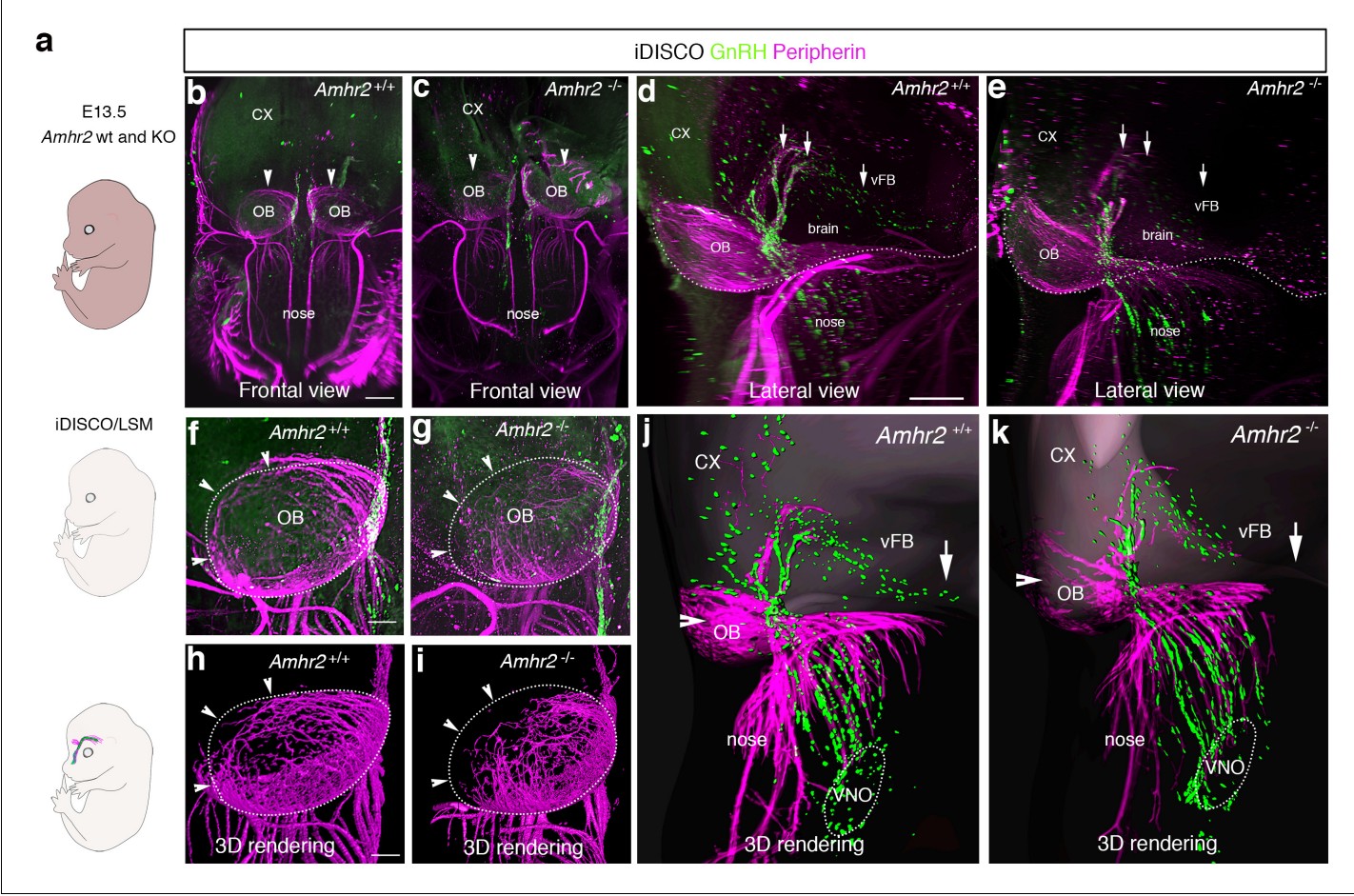

**Figure 3.** GnRH migration and olfactory innervation are perturbed in *Amhr2⁻/⁻* mice. (a) Schematic representation depicting whole-body iDISCO experiments in E13.5 *Amhr2⁺/⁺* and *Amhr2⁻/⁻* embryos. E13.5 embryos (*n* = 2 per genotype) were immunolabelled for Peripherin and GnRH, rendered optically transparent using iDISCO and imaged with a light-sheet microscope (LSM). (b, c) Frontal projection of the embryo heads, arrowheads indicate noticeable differences in Peripherin-positive fibers innervating the olfactory bulb (OB). Lateral projection views (d, e) showing defective GnRH migration and terminal nerve projections to the ventral forebrain (vFB, arrows). (f, g) Higher magnification photomicrographs depicting olfactory axon innervations of the right OB shown in b and c. Dotted circles define the anatomical border of the OB. (h, i) 3D rendering of figures in f and g. Arrowheads indicate observed differences in olfactory axon innervation between *Amhr2⁺/⁺* and *Amhr2⁻/⁻* embryos. (j, k) 3D rendering of peripherin and GnRH staining observed from a lateral projection in a representative *Amhr2⁺/⁺* and *Amhr2⁻/⁻* embryo. Cx: cortex; VNO: vomeronasal organ. Scale bars: (b) 400 µm; (d) 300 µm; (f) 130 µm.

DOI: https://doi.org/10.7554/eLife.47198.006

The following video is available for figure 3:

**Figure 3—video 1.** Light-sheet fluorescence microscopy video of solvent-cleared E13.5 *Amhr2⁺/⁺* and *Amhr2⁻/⁻* embryos immunostained *in toto* for GnRH (green) and peripherin (red).

DOI: https://doi.org/10.7554/eLife.47198.007

compared to wild-type littermates. This is suggestive of an alteration in the hypothalamic network activity in *Amhr2* transgenic animals.

Finally, we tested the fertility of *Amhr2* transgenic female and male mice by performing a constant breeding protocol over three months. We paired either *Amhr2⁺/⁺* sexually experienced males with females belonging to the three different genotypes and, inversely, we paired *Amhr2⁺/⁺* females with *Amhr2⁺/⁻* or *Amhr2⁻/⁻* males (*Figure 4n*). We found a significant impairment of fertility in both *Amhr2⁻/⁻* and *Amhr2⁺/⁻* females, as indicated by fewer litters per months, by fewer pups per litter and by a significant delay in the first litter after pairing as compared to *Amhr2⁺/⁺* females (*Figure 4n*). Heterozygous females displayed an intermediate phenotype between *Amhr2⁺/⁺* and

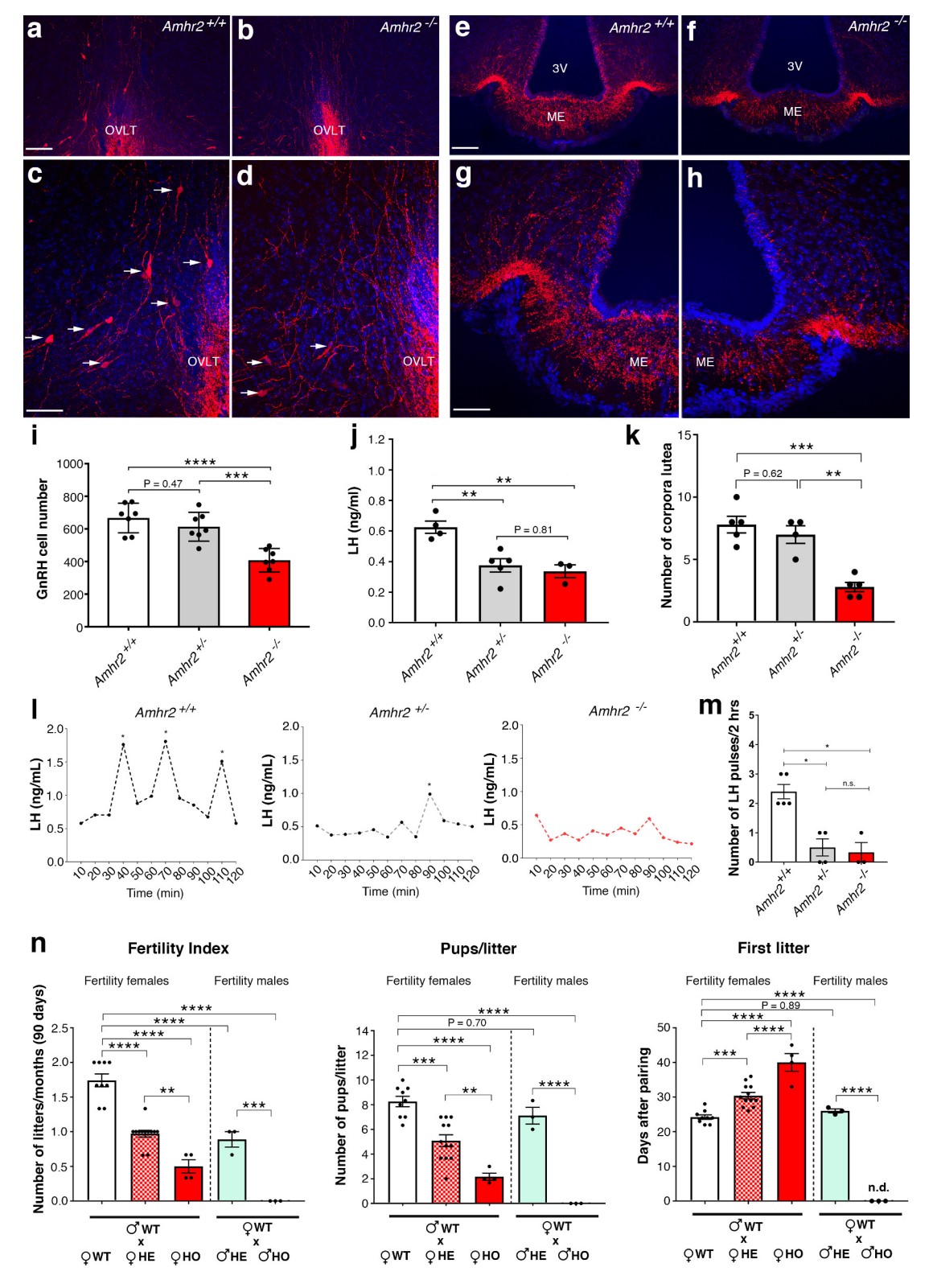

**Figure 4.** *Amhr2* mutant mice show reduced GnRH cell number and impaired LH secretion and fertility. (**a–h**) Immunolabelling of GnRH (red staining) in adult wild type and *Amhr2*-/- adult female mice (P90–P120). The majority of GnRH cell bodies are located at the level of the organum vasculosum laminae terminalis (OVLT) in both *Amhr2* +/+ and *Amhr2* -/- mice, (arrows (**c, d**)). (**e–h**) GnRH fiber projections at the level of median eminence. (**i**) Total mean GnRH population in *Amhr2*+/+, *Amhr2*+/- and *Amhr2*-/- adult female mice brains (3–4 months old). Comparisons between groups were

*Figure 4 continued on next page*

Figure 4 continued

performed using one-way ANOVA followed by Tukey's post hoc test ($n = 7$ for all groups, $F_{2,18}$ <0.0001; $Amhr2^{+/+}$ vs $Amhr2^{+/-}$ $P = 0.4716$; WT vs. $Amhr2^{-/-}$p<0.0001, $Amhr2^{+/-}$ vs $Amhr2^{-/-}$p = 0.0007). (j) Plasma LH levels in adult mature (4–6 months old) diestrous females ($Amhr2^{+/+}$, $n = 4$; $Amhr2^{+/-}$, $n = 5$; $Amhr2^{-/-}$ $n = 3$). Statistical analysis was performed by one-way ANOVA ($F_{2,9}$ = 12.64, p = 0.0024) followed by Tukey's multiple comparison post hoc test ($Amhr2^{+/+}$ vs. $Amhr2^{+/-}$ $P = 0.005$; $Amhr2^{+/+}$ vs. $Amhr2^{-/-}$p = 0.046, $Amhr2^{+/-}$ vs. $Amhr2^{-/-}$p = 0.8164). (k) Quantitative analyses of the mean number of corpora lutea (CL) in $Amhr2^{+/+}$ ($n = 5$), $Amhr2^{+/-}$ ($n = 4$) and $Amhr2^{-/-}$ ($n = 5$) adult ovaries (4–6 months old). Statistical significance between groups was assessed using one-way ANOVA ($F_{2,11}$ = 22.11, p = 0.0001) followed by Tukey's multiple comparison *post hoc* test ($Amhr2^{+/+}$ vs. $Amhr2^{+/-}$ $P = 0.6259$; $Amhr2^{+/+}$ vs. $Amhr2^{-/-}$p = 0.0002 and $Amhr2^{+/-}$ vs. $Amhr2^{-/-}$p = 0.0012). (l) Representative graphs for LH pulsatility in female dioestrous adult mice of the corresponding genotype. Asterisks indicate the number of LH pulses per 2 hr interval. (m) Number of LH pulses in adult (P60) diestrous females ($Amhr2^{+/+}$, $n = 5$; $Amhr2^{+/-}$, $n = 4$; $Amhr2^{-/-}$ $n = 3$). Statistical analysis was performed by non-parametric Kruskal-Wallis test p = 0.0028 ($Amhr2^{+/+}$ vs. $Amhr2^{+/-}$ $P = 0.041$; $Amhr2^{+/+}$ vs. $Amhr2^{-/-}$p = 0.038 and $Amhr2^{+/-}$ vs. $Amhr2^{-/-}$p>0.999). (n) Bar graphs illustrating the results of the constant mating protocol performed over 90 days on the following groups: (♀$Amhr2^{+/+}$ x ♂$Amhr2^{+/+}$, $n = 9$; ♀$Amhr2^{+/-}$ x ♂$Amhr2^{+/+}$, $n = 12$; ♀$Amhr2^{-/-}$ x ♂$Amhr2^{+/+}$, $n = 4$; ♀$Amhr2^{+/+}$ x ♂$Amhr2^{+/-}$, $n = 3$; ♀$Amhr2^{+/+}$ x ♂$Amhr2^{+/-}$, $n = 3$. Female and male mice were 4–6 months-old). Comparisons between groups were performed using one-way ANOVA (fertility index, $F_{4,26}$ = 51.47, p<0.0001; first litter, $F_{4,26}$ = 88.82, p<0.0001; pups per litter, $F_{4,26}$ = 29.67 $P$<0.0001) followed by Tukey's multiple comparison post hoc test, *p<0.05; **p<0.005; ***p<0.0005; ****p<0.0001. Each cluster of data points represents a different mouse. Data were combined from three independent experiments. Throughout the figure, data are displayed as mean ± s.e.m. *p<0.05; **p<0.005; ***p<0.0005; ****p<0.0001. Scale bars: (a, b, e, f) 100 μm; (c, d, g, h) 50 μm.

DOI: https://doi.org/10.7554/eLife.47198.008

The following source data and figure supplements are available for figure 4:

**Source data 1.** This spreadsheet contains the values used to generate the bar plots shown in *Figure 4i j, k, m, n*.
DOI: https://doi.org/10.7554/eLife.47198.011
**Figure supplement 1.** GnRH cell number in *Amhr2* wild-type and knock-out animals as a function of sex.
DOI: https://doi.org/10.7554/eLife.47198.009
**Figure supplement 1—source data 1.** This spreadsheet contains the values used to generate the bar plots shown in *Figure 4—figure supplement 1*.
DOI: https://doi.org/10.7554/eLife.47198.010

$Amhr2^{-/-}$ mice, since all fertility measurements revealed statistically significant differences between $Amhr2^{+/-}$ and $Amhr2^{+/+}$ mice and between $Amhr2^{+/-}$ and $Amhr2^{-/-}$ mice (*Figure 4n*).

In heterozygous males, only the number of litters/90 days was found to be significantly reduced as compared to $Amhr2^{+/+}$ males (*Figure 4n*). $Amhr2^{-/-}$ males are completely infertile (*Figure 4n*), in agreement with previous reports showing that inactivation of *Amhr2* or *Amh* in humans and mice leads to persistent Müllerian duct syndrome (*Imbeaud et al., 1995*; *Mishina et al., 1996*).

Taken together, these data support the physiological relevance of Amh signaling both in the development and homeostasis of the hypothalamic-pituitary-gonadal axis.

## AMH increases GN11 cell migration via the Amhr2/Bmpr1b signaling pathway

The manipulation of the GnRH migratory system and functional experimentation on these neurons have been challenging because of their limited number (800 in rodents) and anatomical dispersal along their migratory route. The generation of immortalized GnRH neurons (GN11 and GT1-7 cell lines [*Radovick et al., 1991*; *Mellon et al., 1990*]) has permitted the study of immature migratory (GN11 cells), and mature post-migratory (GT1-7 cells) GnRH neurons, respectively.

To assess whether immortalized GnRH cell lines retain expression of *Amh* and *Amh* receptors, RT-PCR analysis was performed (*Figure 5a*). Our data show that both GN11 and GT1-7 cells express *Amh* and *Amhr2*, even though the transcript levels were significantly higher in GT1-7 cells as compared to GN11 cells (*Figure 5a*). These data are consistent with our current (*Figure 1c*) and previous findings (*Cimino et al., 2016*) obtained from GnRH sorted cells. As for the Amh-type one receptors, both cell lines express *Acvr1* and *Bmpr1a*, with GT1-7 cells displaying higher levels of expression compared to GN1 cells (*Figure 5a*). Interestingly, GN11 cells, but not GT1-7 cells, express *Bmpr1b* (*Figure 5a*), indicating that Amhr2/Bmpr1b signaling maybe a putative hallmark of migratory GnRH neurons. These results point to the GN11 cell line as an appropriate in vitro model to test the functional role of Amh on cell motility.

Activation of the MAPK pathway (phosphorylation of ERK1/2) has been previously associated with increased GN11 cell motility (*Messina et al., 2011*; *Hanchate et al., 2012*). Here, we found that AMH, at concentrations previously reported to be functional in other cell lines (*Garrel et al., 2016*), significantly increased the phosphorylation of ERK1/2 in GN11 cells in a dose- and time-dependent

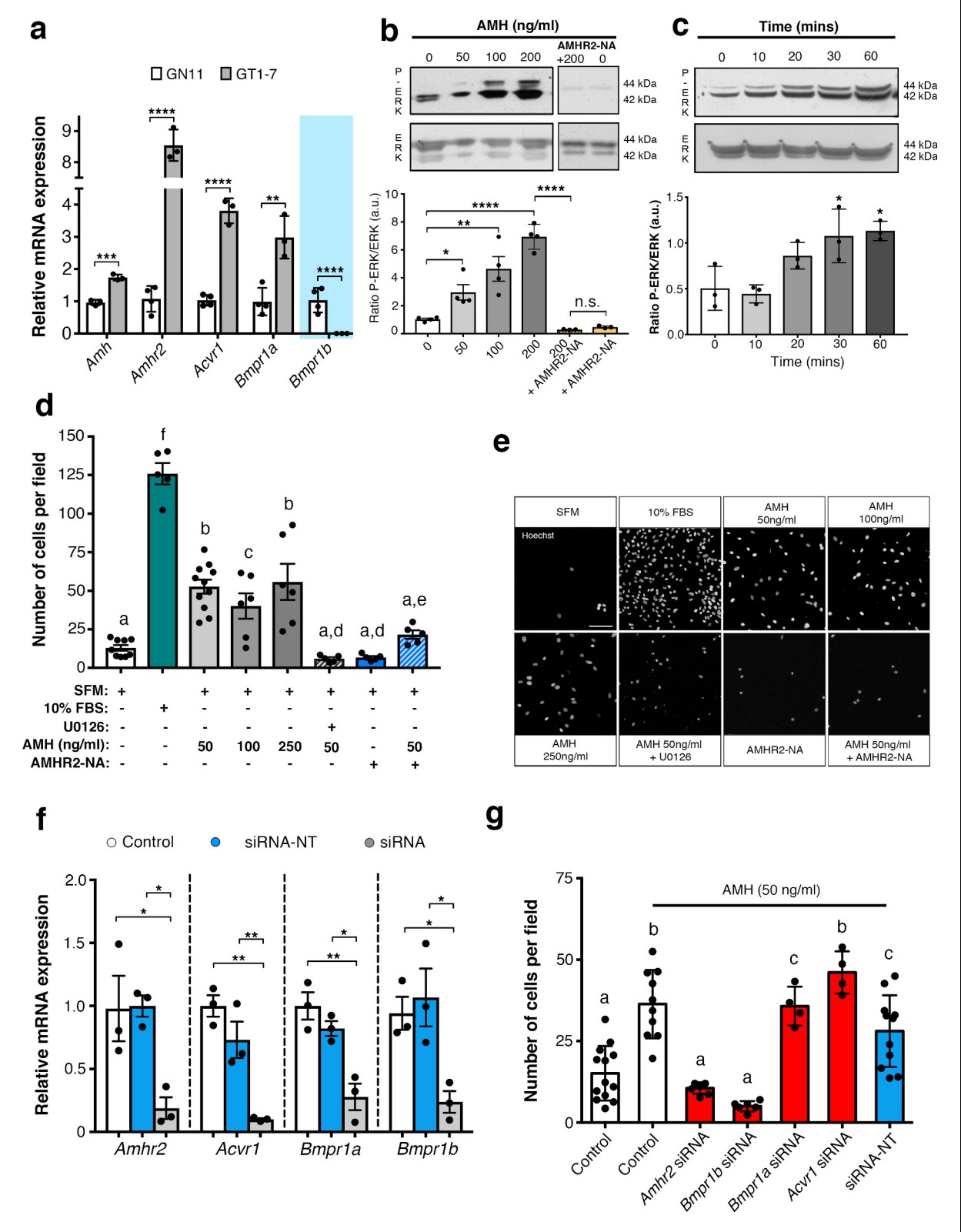

**Figure 5.** AMH promotes GnRH cell motility via Amhr2 and Bmpr1b signaling. (a) Quantitative RT-PCR analysis for *Amh, Amhr2, Acvr1* (Activin Receptor1; ALK2), *Bmpr1a* (Bone Morphogenetic Protein Receptor1a; ALK3) and *Bmpr1b* (Bone Morphogenetic Protein Receptor1b; ALK6) mRNA in GN11 ($n$ = 4) and GT1-7 ($n$ = 3) cells. Comparisons between treatment groups were performed using unpaired two-tailed Student's $t$ test (*Amh* $t_5$ = 1.139, p = 0.0004; *Amhr2* $t_5$ = 1.6, p<0.0001; *Acvr1* $t_5$ = 5.044, p<0.0001); *Bmpr1a* $t_5$ = 2.374, p<0.0044. (b) Representative western blot showing P-ERK1/
*Figure 5 continued on next page*

*Figure 5 continued*

2 and total ERK1/2 in cell lysates of GN11 cells stimulated with indicated doses of AMH ($n$ = 4). Right boxed figure is a representative blot showing P-ERK1/2 and total ERK1/2 in cell lysates of GN11 cells stimulated with anti-Amhr2 neutralizing antibody with or without 200 ng/ml of AMH (Amhr2-NA, $n$ = 3 per condition). Bar graph illustrates the mean ratio P-ERK1/2 over total ERK1/2 ($n$ = 4 for all except AMHR2-NA and AMHR2-NA + AMH 200 ng/ml, $n$ = 3). Comparisons between treatment groups were performed using a two-way ANOVA ($F_{6,19}$ = 29.11, p<0.0001; followed by Holm-Šídák's multiple comparison post hoc test. Adjusted p values: 0 vs. 50 = 0.0461, 0 vs 100 = 0.0003, 0 vs 200 =< 0.0001, 200 vs AMHR2-NA + 200 =< 0.0001, 0 vs AMHR2-NA => 0.9999). (c) Representative western blot showing P-ERK1/2 and total ERK1/2 in cell lysates of GN11 cells stimulated with 50 ng/ml of AMH for the indicated times (minutes: min). Bar graph illustrates the mean ratio P-ERK1/2 over total ERK1/2 ($n$ = 3 for all). Comparisons between treatment groups were performed using a one-way ANOVA ($F_{4,10}$ = 8.171, followed by Tukey's multiple comparison post hoc test. Adjusted p values: 0 vs. 10 = 0.9945, 0 vs. 20 = 0.2333, 0 vs. 30 = 0.0292, 0 vs. 60 = 0.0170). (d) Schematic representation on top of the graph bar illustrates the transwell assay used to assess cell motility in d, e, g, whereby AMH was placed on the top and lower chamber. Bar graph illustrates the mean number of migrated GN11 cells stimulated with serum free medium (SFM, basal conditions, $n$ = 9), with 10% fetal bovine serum (FBS, strong inducer of cell motility, $n$ = 5), or with the indicated doses of AMH with or without the MAPK Kinase inhibitor, U0126 (AMH 50 ng/ml $n$ = 11, AMH 100 ng/ml $n$ = 6, AMH 250 ng/ml $n$ = 6, AMH 50 ng/ml + U0126 $n$=5), or with Amhr2-NA with or without AMH 50 ng/ml ($n$ = 5). One-way ANOVA, $F_{7,44}$ = 38.48, followed by Tukey's multiple comparison post hoc test. (a): not significantly different from a groups (p>0.05); b: significantly different from a) groups (p<0.0001); c: SFM vs AMH 100 ng/ml, p<0.05; d: significantly different from b groups (p<0.001); e: AMH 50 ng/ml vs AMH 50 ng/ml + AMHR2 NA, p<0.05; f: significantly different from every other group (p<0.0001). (e) Representative photomicrographs showing Hoechst nuclear staining of the migrated GN11 cells after the different treatments, scale bar = 100 μm. (f) Real-time PCR analysis for *Amhr2, Acvr1, Bmpr1a* and *Bmpr1b* mRNA expression in untrasfected GN11 cells (Control) or in GN11 cells transfected with siRNAs targeting Amh receptors or with a non-targeting siRNA (siRNA-NT) ($n$ = 3). Bar graph illustrates the mean ± s.e.m; one-way ANOVA followed by Tukey's post hoc comparison test (Amhr2 $F_{2,6}$ = 7.861, Control vs siRNA p = 0.0339, Control vs siRNA-NT p = 0.9958, siRNA vs siRNA-NT p = 0.0305; Acvr1 $F_{2,6}$ = 22.73, Control vs siRNA p = 0.0015, Control vs siRNA-NT p = 0.2016, siRNA vs siRNA-NT p = 0.0088; Bmpr1a $F_{2,6}$ = 16.16, Control vs siRNA p = 0.0038, Control vs siRNA-NT p = 0.4206, siRNA vs siRNA-NT p = 0.0149; Bmpr1b $F_{2,6}$ $P$ = 7.777, Control vs siRNA p = 0.0478, Control vs siRNA-NT p = 0.8489, siRNA vs siRNA-NT p = 0.0247). (g) Transwell assay was performed on GN11 cells transfected or not with indicated siRNAs and stimulated with or without AMH (50 ng/ml). Bar graph illustrates the mean number of migrated GN11 cells (Control, SFM $n$ = 13, Control +AMH 50 ng/ml $n$ = 10, *Amhr2* siRNA +AMH 50 ng/ml $n$ = 7, *Acvr1* siRNA +AMH 50 ng/ml $n$ = 4, *Bmpr1a* siRNA +AMH 50 ng/ml $n$ = 4, *Bmpr1b* siRNA +AMH 50 ng/ml $n$ = 6, siRNA-NT +AMH 50 ng/ml $n$ = 11). Comparisons between treatment groups were performed using a one-way ANOVA followed by Tukey's post hoc comparison test ($F_{6,48}$ = 20.99, a not significantly different from other groups denoted a, p>0.05; b significantly different from groups denoted a, p<0.0001; c significantly different from groups denoted a), p<0.05. Throughout the figure, data were combined from three independent experiments and displayed as mean ± s.e.m. *p<0.05; **p<0.005; ***p<0.0005; ****p<0.0001.

DOI: https://doi.org/10.7554/eLife.47198.012

The following source data is available for figure 5:

**Source data 1.** This spreadsheet contains the values used to generate the bar plots shown in *Figure 5a, b, c, d, f and g*.

DOI: https://doi.org/10.7554/eLife.47198.013

manner (*Figure 5b,c*). The AMH-dependent activation of MAPK pathway was prevented by the pharmacological blockage of Amhr2 (AMHR2 neutralizing antibody; AMHR2-NA; *Figure 5b*).

Using transwell assays, we showed that recombinant human AMH was able to significantly increase the motility of GN11 cells at all tested doses (50 ng/ml; 100 ng/ml; 250 ng/ml) compared to controls (serum-free medium, SFM; *Figure 5d*). In agreement with our biochemical results (*Figure 5b,c*), the AMH-dependent induction of cell motility was prevented by the selective pharmacological antagonist of MAPK pathway (U0126 inhibits MEKK1, the upstream activator of ERK) and by AMHR2-NA (*Figure 5d,e*).

We next investigated which receptor complex was required to mediate the AMH-dependent cell migration in GN11 cells. This was achieved by targeted knockdown of Amh receptors through a small interfering RNA (siRNA) strategy. GN11 cells were transfected with a pool of siRNAs specific to mouse *Amhr2, Acvr1, Bmpr1a, Bmpr1b,* or with a pool of nontargeting siRNAs (siRNA-NT). Silencing efficiency was assessed analyzing gene expression levels in untransfected cells (Control) versus GN11 cells transfected with the *Amh* receptors targeted siRNAs and siRNA-NT (*Figure 5f*).

Knockdown of individual *Amh* receptors led to distinct motility responses of GN11 cells to AMH stimulation (*Figure 5g*). Transfection with the siRNA-*Acvr1*, siRNA-*Bmpr1a* or siRNA-NC RNA did not affect the GN11 response to AMH treatment (*Figure 5g*). In contrast, knockdown of *Amhr2* and *Bmpr1b* resulted in significantly reduced GN11 cell motility in response to AMH as compared to the control groups (mock and siRNA-NC transfected cells; *Figure 5g*).

These data show that AMH promotes GN11 cell motility through the Amhr2/Bmpr1b receptor complex and activation of the MAPK intracellular pathway.

**Table 1.** Summary of heterozygous *AMH* or *AMHR2* mutations identified in patients with congenital hypogonadotropic hypogonadism.
cDNA and protein changes are based on reference cDNA sequence NM_000479.4 (*AMH*) and NM_020547.3 (*AMHR2*). Functional validation of the mutants has been performed in vitro evaluating AMH secretion in COS-7 cells, cell motility in GN11 cells, and measuring GnRH secretion in GT1-7 cells for nCHH-associated mutants. CHH, congenital hypogonadotropic hypogonadism; nCHH, normosmic CHH; KS, Kallmann syndrome; Sex: F: female; M: male; Inheritance: F: familial, S: sporadic; Puberty: A: absent puberty, P: partial puberty. MAF, minor allele frequency; ↓, decreased; NS, not significant; NA, not applicable.

| Gene | Family | Subject | Diagnosis | Sex | Inheritance | Puberty | Associated phenotypes | dbSNP number | Nucleotide change | Amino acid change | MAF (%) gnomAD MaxPop | In vitro studies | | |
|---|---|---|---|---|---|---|---|---|---|---|---|---|---|---|
| | | | | | | | | | | | | Released AMH (COS-7 cells) | Cell motility (GN11 cells) | GnRH secretion (GT1-7 cells) |
| AMH | 1 | II-1 | nCHH | M | F | P | High-arched palate Deviated nasal septum Hyperlaxity | rs2002 26465 | c.295A > T | p.Thr99Ser | 0.044 | ↓↓ | ↓↓ | ↓↓ |
| | 2 | II-1 | KS | M | S | A | Cryptochidism | rs3705 32523 | c.451C > T | p.Pro 151Ser | 0.011 | ↓↓ | ↓↓ | |
| | 3 | II-2 | KS | F | F | P | Osteoporosis Scoliosis | rs7525 74731 | c.714C > A | p.Asp 238Glu | 0.006 | ↓↓ | ↓↓ | |
| AMHR2 | 4 | II-1 | nCHH | F | S | A | Osteoporosis | rs7647 61319 | c.1330_ 1356del | p.Gly445_ Leu453del | 0.093 | ↓↓ | ↓↓ | ↓↓ |

DOI: https://doi.org/10.7554/eLife.47198.015

## CHH patients harbor heterozygous *AMH* and *AMHR2* mutations

In this study, we performed whole exome sequencing in 75 KS and 61 normosmic CHH (nCHH) probands who did not harbor pathogenic mutations in known CHH genes, and identified in three probands from European descent heterozygous missense mutations in *AMH* (*Table 1*, *Figure 6a,b*). These mutations (p.Thr99Ser, p.Pro151Ser, and p.Asp238Glu) lie in the N-terminal pro-protein domain (*Figure 6a*), and all affected amino acids were highly conserved across species (*Figure 6c*). Additionally, one female with normosmic CHH (nCHH) harbors a heterozygous in-frame 27-nucleotide deletion in *AMHR2*. This p.Gly445_Leu453del deletion lies within the catalytic intracellular serine/threonine domain of the receptor (*Figure 6d–6f*).

We observed variable degrees of spontaneous puberty (absent to partial) among the probands carrying an *AMH* or *AMHR2* mutations. Two probands had KS with no other major associated non-reproductive phenotypes (*Table 1* and human case summaries in Materials and methods). All three probands with mutations in *AMH* (Families 1, 2, and 3) have a positive family history for partial phenotypes (e.g. delayed puberty, anosmia), consistent with variable expressivity (*Figure 6b*). The female proband carrying the *AMHR2* deletion (Family 4) has nCHH. Her mother, who did not carry the deletion, exhibited cleft lip/palate with normal reproduction (*Figure 6e*).

## *AMH* and *AMHR2* mutations in CHH are loss-of-function

In order to test the functionality of the *AMH* and *AMHR2* mutants identified in KS and nCHH probands, we first transiently transfected COS-7 cells with plasmids encoding the human *AMH* wild-type (*AMH* WT) or the *AMH* variants and investigated whether the AMH secretory capacity of transfected cells was affected. All three of mutations tested (p.Pro151Ser, p.Asp238Glu, and p.Thr99Ser *AMH* mutants) showed significantly reduced AMH protein secretion in vitro (*Figure 7a* and *Table 1*), as assessed by automated chemoluminescent immunoassay.

To test the impact of *AMH* mutants on immortalized GnRH neurons' cell motility and to determine whether AMH promotes such response through an autocrine mechanism, we performed transwell migration assays on GN11 cells either treated with lipofectamine (mock), or transfected with the *AMH* WT or the *AMH* variants identified in CHH patients (*Figure 7b*). AMH overexpression (*AMH* WT) in GN11 cells significantly increased cell migration by 50% when compared with mock cells (*Figure 7b*). The AMH-dependent induction of cell motility was prevented when the cells where transfected with the mutants identified in KS patients (p.Pro151Ser and p.Asp238Glu) as well as with the mutant found in a nCHH proband (p.Thr99Ser; *Figure 7b* and *Table 1*). Moreover, since the latter *AMH* mutation was found in a male nCHH proband (LH 2.7 U/l; *Table 1*), and because we previously showed that AMH stimulates GnRH and LH secretion in rodents (*Cimino et al., 2016*), we wondered whether this mutant could also negatively impact on GnRH secretion. In order to assess that we used GT1-7 cells that express Amh type-I and type-II receptors (*Figure 5a*) and that display significant GnRH secretory activity (*Mellon et al., 1990*). GT1-7 cells were transfected with either *AMH* WT or p.Thr99Ser *AMH* and conditioned medium was collected 48 hr later for GnRH ELISA measurement. The p.Thr99Ser *AMH* variant significantly reduced GnRH secretion as compared to GT1-7 cells expressing the *AMH* WT (*Figure 7c*).

To functionally test the impact of the *AMHR2* deletion (p.Gly445_Leu453del) and determine whether this variant leads to defective AMH-induced motility, we transfected GN11 cells with a plasmid encoding the h*AMHR2* WT or the *AMHR2* p.Gly445_Leu453del mutant and performed migration assays culturing the cells with SFM alone or supplemented with recombinant human AMH protein (*Figure 7d*). Consistent with the data shown in *Figure 5*, AMH treatment significantly increased cell migration of *AMHR2* WT-transfected cells as compared with SFM (*Figure 7d*). This effect was significantly impaired when GN11 cells were transfected with the *AMHR2* mutant plasmid (*Figure 7d*). Since the *AMHR2* p.Gly445_Leu453del mutant was found in a female nCHH proband (LH <2.0 U/l; *Table 1*; human case summaries in Materials and methods), we also assessed whether AMH treatment increased GnRH release in GT1-7 cells expressing either *AMHR2* WT or the p.Gly445_Leu453del mutant (*Figure 7e*). These experiments revealed that AMH (50 ng/ml) stimulates GnRH secretion into medium of GnRH cells expressing the *AMHR2* WT, whereas introduction of the *AMHR2* mutant variant into GT1-7 cells significantly reduced the AMH-dependent GnRH secretion as compared to control conditions (*Figure 7e*).

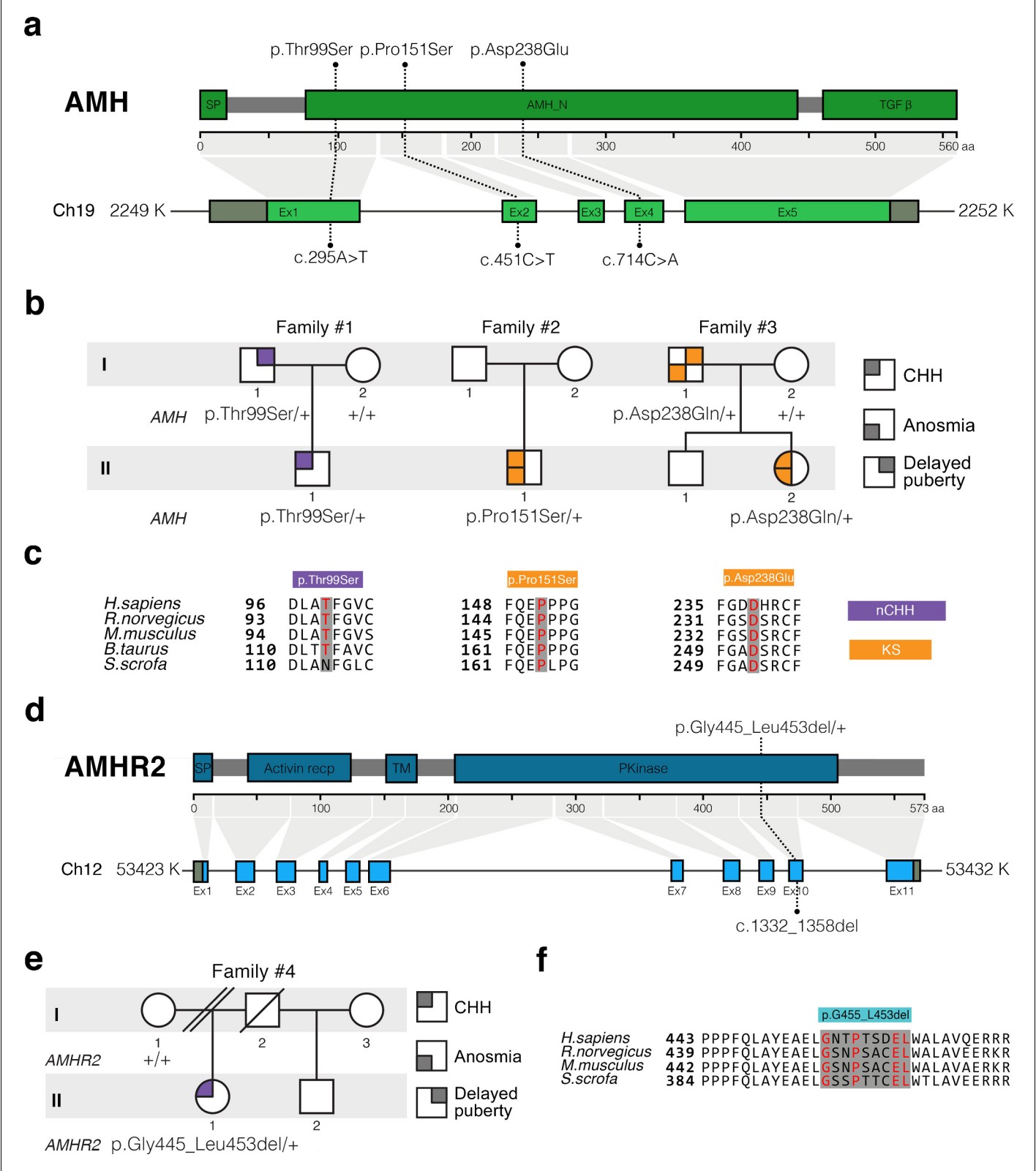

**Figure 6.** *AMH* and *AMHR2* heterozygous mutations in CHH probands. (**a**) Schematic illustration of *AMH* mutations in nCHH and KS probands. (**b**) Pedigrees of patients harboring *AMH* mutations. Circles denote females, squares denote males. The phenotype interpretation is explained in the square legend on the top of the figure. (**c**) The *AMH* mutations affect evolutionarily conserved amino acid residues. Alignment of partial protein sequences of *AMH* orthologs showing in red text the amino acid residues evolutionarily conserved. Purple highlights correspond to variants identified

*Figure 6 continued on next page*

*Figure 6 continued*

in nCHH probands and orange highlights correspond to variants identified in the KS cohort. (**d**) Schematic of the AMHR2 signal peptide (SP), activin receptor, transmembrane and kinase functional domains along with the p.Gly445_Leu453del variant identified in the cohort. This deletion lies within the catalytic intracellular serine/threonine kinase domain (PKinase) of the receptor. (**e**) Pedigree of the patient harboring the deletion in *AMHR2*. Circles denote females, squares denote males, double diagonal lines indicate divorce, single diagonal line indicates death. The phenotype interpretation is explained in the square legend on the top of the figure. (**f**) Alignment of partial protein sequences of mammalian AMHR2 orthologs flanking the deletion site.

DOI: https://doi.org/10.7554/eLife.47198.014

Finally, to evaluate the structural impact of the AMHR2 deletion (p.Gly445_Leu453del), a three-dimensional structural model of the corresponding mutated catalytic intracellular serine/threonine domain of the receptor was generated (DEL), as previously described (*Belville et al., 2009*). The model of the WT AMHR2 kinase domain presents a general fold of a two-domain kinase, with an N-lobe mainly composed of a five-stranded β-sheet and a mostly α-helical C-lobe. The deleted residues are located at the top of the C-lobe and are part of the αG helix and its preceding loop (*Figure 7f*). In both WT and DEL, the overall protein structure remains stable (*Figure 7—figure supplement 1*). Analysis of the interactions established in this zone reveals differences between the *AMHR2*-WT and the *AMHR2* p.Gly445_Leu453del mutant models. In the WT model, the structure is stabilized by hydrophobic interactions involving Leu444, Leu456 and Leu453, as well as by the hydrogen bonds Arg462-Glu443, Arg463-Tyr440 and Gln446-Glu441. For the *AMHR2* p.Gly445_Leu453-del mutant model, the structure is stabilized mainly by hydrogen bonds: Arg462-Glu443, Arg462-Glu460 and Arg463-Tyr440 (*Figure 7—figure supplement 2*). The main structural fluctuations are observed in the loop regions of the proteins (*Figure 7g–i*). Comparison of *AMHR2* WT and *AMHR2* p.Gly445_Leu453del mutant simulations (*Figure 7g–i*) suggests there may be some differences in the dynamic behavior of some of these flexible regions, including the kinase activation loop. In summary, although *AMHR2* mutant tertiary structure is expected to be folded in a similar manner to that of the WT species, it is possible that the deletion results in some alterations in the intracellular signaling.

Taken together, these in vitro results confirm that the identified AMH and AMHR2 mutants are loss-of-function, supporting the role of AMH/AMHR2 signaling in GnRH neuronal migration and GnRH secretion and thus pointing toward a potential contribution of these variants to the pathogenesis of CHH.

## Discussion

Originally identified in the mesenchyme of Müllerian ducts and in gonads (*Josso et al., 1998*), Amh and Amhr2 were subsequently documented in several other organs, including the brain (*Lebeurrier et al., 2008*; *Wang et al., 2009*; *Cimino et al., 2016*; *Wang et al., 2005*) and the pituitary (*Garrel et al., 2016*), suggesting that Amh biological effects could be much broader than initially thought.

We recently reported Amhr2 expression in migratory GnRH neurons and along olfactory axons, both in mice and human fetuses (*Cimino et al., 2016*). In this study, we showed that GnRH neurons express Amh during fetal development and that this expression is retained both in rodents and humans. Our in vivo and in vitro analyses show that Amh signaling regulates migration of GnRH neurons toward the brain through an autocrine mechanism. This is strongly supported by our in vivo and in vitro data showing Amh expression in GnRH neurons and by the reduction in cell motility detected in GN11 cells when transfected with the h*AMH* CHH variants. Moreover, in this study, we showed that Amh acts as a promotility factor for GnRH neurons by signaling via Amhr2/Bmpr1b and activation of the MAPK pathway.

The animal experiments revealed that both acute neutralization of Amhr2 and genetic invalidation of this receptor lead to a strong accumulation of GnRH cells in the nasal region with defects in both the olfactory targeting to the OBs and alterations in the intracranial projections of the VNN/TN; defects that strongly resemble the phenotype previously described in histological analyses of KS

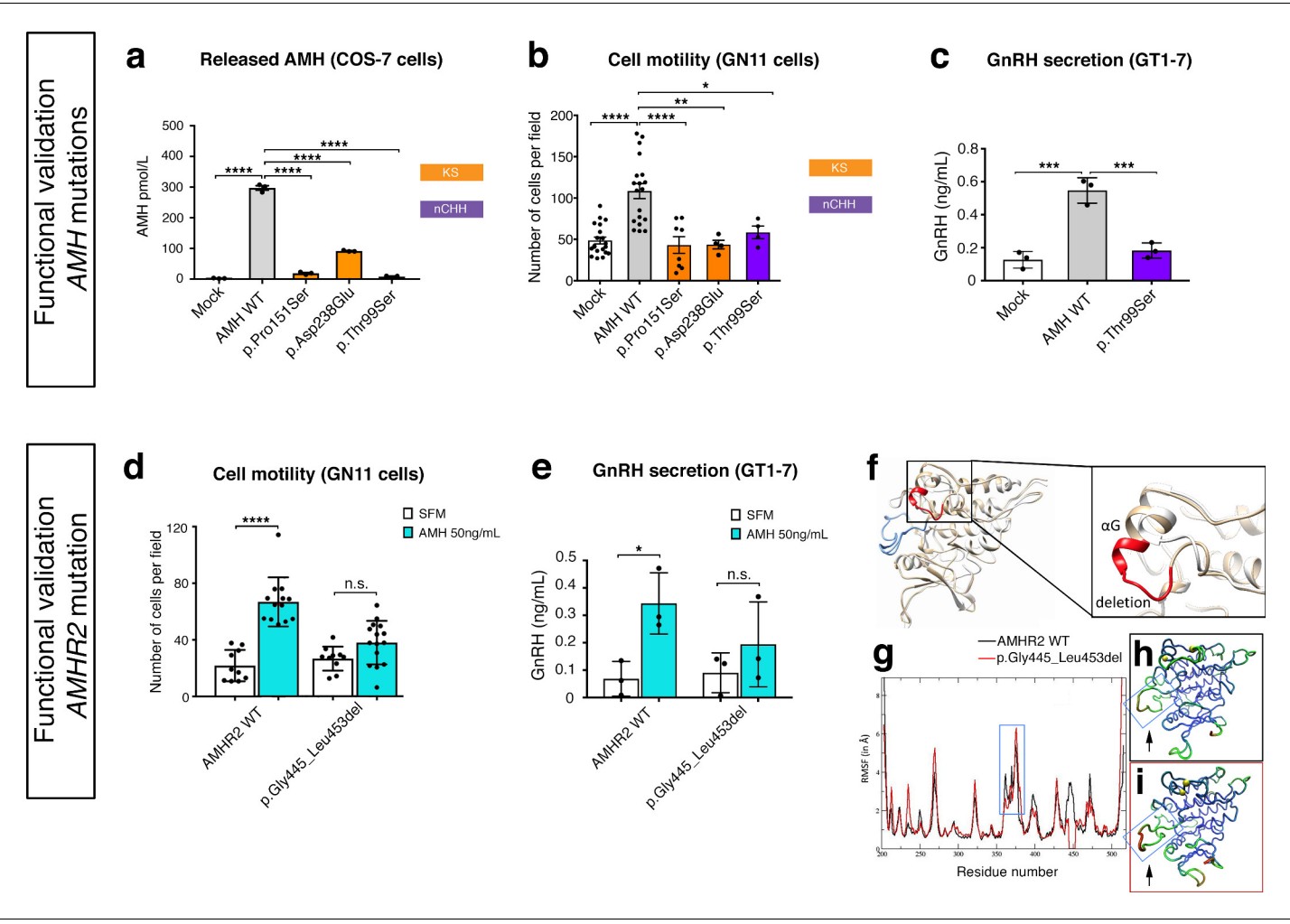

**Figure 7.** Functional validation of AMH variants. (a) AMH released in the medium of COS-7 cells transiently transfected either with lipofectamine alone (mock), or with a WT AMH or a variant AMH identified in CHH and KS probands. Bar graph illustrates the mean amount of AMH secreted in the conditioned medium of transfected COS-7 cells ($n$ = 3 independent experiments per condition). Comparisons between treatment groups were performed using a one-way ANOVA followed by Tukey's post hoc comparison test ($F_{4,10}$ = 1193, Mock vs AMH WT p<0.0001, AMH WT vs p.Pro151Ser p<0.0001, AMH WT vs p.Asp238Glu p<0.0001, AMH WT vs p.Thr99Ser p<0.0001). No significant motility difference was detected between Mock, p.Thr99Ser and p.Pro151Ser mutated forms of AMH treatment, p>0.9 for all. (b) Transwell assay was performed on GN11 cells transiently transfected either with lipofectamine alone (mock), or with a WT AMH or a variant AMH identified in CHH and KS probands. Comparisons between treatment groups were performed using a one-way ANOVA followed by Tukey's post hoc comparison test ($F_{4,50}$ = 13.94, Mock vs AMH WT p<0.0001, AMH WT vs p.Pro151Ser p<0.0001, AMH WT vs p.Asp238Glu p<0.0014, AMH WT vs p.Thr99Ser p = 0.0218. No significant motility difference was detected between Mock and mutated forms of AMH treatment, p>0.9 for all. (c) Quantification of GnRH secretion from GT1-7 cells transfected with lipofectamine alone (mock), or with a WT AMH or the p.Pro151Ser AMH variant identified in a nCHH proband. GnRH mean concentration measured in the medium ($n$ = 3, one-way ANOVA: $F_{2,6}$ = 43.84, p = 0.0003; followed by Tukey's multiple comparison post hoc test, mock vs. AMH WT p = 0.0003, mock vs p.Thr99Ser p = 0.5220, AMH WT vs p.Thr99Ser p = 0.0007. (d) Transwell assay was performed on GN11 cells transiently transfected with the AMHR2 plasmid or with the AMHR2 variant and stimulated with either serum-free medium (SFM) or with recombinant AMH (50 ng/ml). Bar graph illustrates the mean number of migrated GN11 cells under different treatment conditions (SFM $n$ = 10 for both WT and mutant AMHR2, AMH 50 ng/ml $n$ = 12 for both WT and mutant AMHR2). Comparisons between treatment groups were performed using two-way ANOVA ($F_{1,43}$ = 16.5 $P$ = 0.0002; followed by Sidak's multiple comparison post hoc test, AMHR2 WT SFM vs AMHR2 WT + AMH 50 ng/ml p<0.0001, p.Gly445_Leu453del SFM vs p. Gly445_Leu453del + AMH 50 ng/ml $P$ = 0.1036). (e) Quantification of GnRH secretion from GT1-7 cells transfected with the same plasmids as in d ($n$ = 3 independent experiments per condition). Experiments were replicated three times with comparable results. Two-way ANOVA, $F_{1,8}$ = 1.927, p<0.02025; followed by Holm-Šídák multiple comparison post hoc test, AMHR2 WT SFM vs AMHR2 WT + AMH 50 ng/ml $P$ = 0.0269, p.Gly445_Leu453del SFM vs p.Gly445_Leu453del + AMH 50 ng/ml $P$ = 0.4652. (f) Initial three-dimensional models of WT and p.Gly445_Leu453del catalytic intracellular serine/ threonine domains of AMHR2. The backbone of the WT and deleted proteins are shown in tan or white cartoon representations, respectively, with the deleted 445–453 residues colored in red. The activation loop is depicted in blue. (g–i) Root-mean-square fluctuations (RMSF) of the $C_\alpha$ atoms along the simulations for the AMHR2 WT and the p.Gly445_Leu453del models. (g) RMSF (in Å) for the WT (black line) and the p.Gly445_Leu453del models (red

*Figure 7 continued on next page*

*Figure 7 continued*
line, being the average over the three 100 ns simulations) are given for each residue of the protein. For a better comparison, residue numbers were kept the same for both models. Molecular representation of the WT (h) and p.Gly445_Leu453del (i) models colored by RMSF: the blue-green-red scale corresponds to low-medium-high RMSF values. The yellow spheres indicate the first residues after the p.Gly445_Leu453del deletion. The activation loop region is highlighted inside a blue frame (arrows).
DOI: https://doi.org/10.7554/eLife.47198.016
The following source data and figure supplements are available for figure 7:

**Source data 1.** This spreadsheet contains the values used to generate the bar plots shown in *Figure 7a–e*.
DOI: https://doi.org/10.7554/eLife.47198.019
**Figure supplement 1.** Root-mean-square deviations (RMSD) of the AMHR2 protein backbone along the simulations.
DOI: https://doi.org/10.7554/eLife.47198.017
**Figure supplement 2.** Molecular representation of main interactions stabilizing the zone around the AMHR2 deletion.
DOI: https://doi.org/10.7554/eLife.47198.018

human fetuses (*Schwanzel-Fukuda et al., 1996*; *Teixeira et al., 2010*). Since Amh is only produced by GnRH neurons in the fetal brain and because the axonal scaffold of GnRH neurons express Amhr2, we hypothesize that Amh signaling contributes to the correct development of the ventral branch of the vomeronasal/terminal nerves in the brain through a paracrine mechanism. Mono-allelic inactivation of *Amhr2* in mice is not sufficient to significantly alter GnRH neuronal migration, as indicated by the normal number of GnRH neurons observed in adult *Amhr2* heterozygous brains. On the other hand, the presence of only one *Amhr2*-null allele is sufficient to trigger significant impairments of LH secretion, LH pulsatlity and fertility in adult female mice. In heterozygous males, only the number of litters/90 days was found to be significantly reduced as compared to *Amhr2*$^{+/+}$ males, suggesting that sexual behavior but not fecundity is likely altered in *Amhr2*$^{+/-}$ males.

Bi-allelic inactivation of *Amhr2* in mice results instead into a significant reduction of the GnRH cell population in the brains of adult *Amhr2*-null mice of both sexes as compared to wild-type animals. Since male and female homozygous adult animals showed a comparable GnRH cell loss, it is likely that the GnRH migratory defect observed in *Amhr2*-deficient embryos is independent of the genetic sex of the animals.

We speculate that Amh/Amhr2 signaling can regulate, respectively, GnRH migration, during embryonic development, and GnRH/LH secretion postnatally. The latter point is also supported by our in vitro experiments showing AMH-induced GnRH secretion in GT1-7 cells.

Homozygous *Amhr2* female mice have a more pronounced phenotype than heterozygous animals, as they combine developmental defects in GnRH migration with severely impaired ovulation and fertility in adulthood. This strong phenotype could be the consequence of a lack of a broad spectrum of actions of Amh at different prenatal and postnatal stages, impacting the GnRH neuronal migration and the GnRH secretion, respectively, or gonadotrope function (*Garrel et al., 2016*; *Garrel et al., 2019*). Moreover, since Amh is known to be expressed by granulosa cells in the ovaries (*Vigier et al., 1984*) and to regulate folliculogenesis (*Durlinger et al., 2001*; *Durlinger et al., 1999*), it is likely that part of the reproductive phenotype of *Amhr2*$^{-/-}$ mice is also due in part to the lack of ovarian Amh. Dissecting the specific contribution of ovarian versus cerebral Amh in the control of fertility would only be possible using a neuronal specific knockout of *Amhr2*, which is not available yet. However, our previous (*Cimino et al., 2016*) and current findings, showing that Amh directly increases GnRH and LH secretion, support a role for Amh in the neuroendocrine regulation of fertility in physiological and pathological conditions.

Our study identified heterozygous mutations in CHH probands that affect highly conserved amino acids of AMH or its exclusive binding receptor, AMHR2. The AMH mutants display defects in AMH release. Since the described proAMH cleavage sites (*Mamsen et al., 2015*; *Pankhurst and McLennan, 2013*) are not located in close proximity of these mutations, it is unlikely that they impinge on the cleavage of the proAMH. The decreased AMH secretion might rather result from altered protein trafficking leading to accumulation of the mutants AMH in the endoplasmic reticulum (ER) and defective release. The AMH mutants identified in KS probands also significantly reduced GN11 migration compared to the wild-type AMH. Interestingly, the p.Thr99Ser *AMH* mutation found

in a nCHH proband compromises both GnRH cell motility in GN11 cells as well as GnRH secretion in GT1-7 cells. It is plausible that AMH mutants cause dominant negative effects by forming heterodimers with wild-type AMH, thus reducing AMHR2 activation and the downstream biological response. Finally, the AMHR2 p.Gly445_Leu453del mutation identified in a female nCHH proband was also loss-of-function in in vitro migration and GnRH secretion assays.

The heterozygous mutations in *AMH* and *AMHR2* are found in 3% of CHH probands in our cohort (4 out of 136). This is consistent with the genetic landscape of CHH as the majority of known CHH genes have a low mutational prevalence (<5%) (*Boehm et al., 2015*). Variable expressivity was observed in family members carrying the same mutation, consistent with the fact that CHH represents the more severe end of a large spectrum of manifestations. This is a common theme in the genetics of CHH, and factors such as digenic/oligogenic inheritance (*Boehm et al., 2015*; *Sykiotis et al., 2010*; *Pitteloud et al., 2007*), epigenetic regulation or non-genetic contributions likely play important roles. As the current study is limited by the small number of probands harboring *AMH* and *AMHR2* mutations, confirmation in larger CHH cohorts will be necessary to establish the specific contributions of these two genes in the pathogenesis of CHH.

Notably, homozygous or compound heterozygous loss-of-function mutations in *AMH* (OMIM: 600957) or *AMHR2* (OMIM: 600956) cause PMDS in both mice and humans (*Imbeaud et al., 1995*; *Mishina et al., 1996*). PMDS is characterized by the retention of Müllerian duct derivatives (the uterus, fallopian tubes, and upper part of the vagina) in males (*Josso and Clemente, 2003*; *Behringer et al., 1994*; *Mishina et al., 1996*; *Belville et al., 1999*; *Belville et al., 2004*; *Orvis et al., 2008*). Interestingly, we identified two mutations (AMH p.Pro151Ser and AMHR2 p.Gly445_Leu453-del) (*Picard et al., 2017*) in our CHH cohort previously associated with autosomal recessive PMDS. MRI or pelvic ultrasounds were performed in the two male CHH patients harboring *AMH* mutations, including the patient carrying the p.Pro151Ser. No defects in the internal genitalia were identified, consistent with the fact that monoallelic defects in *AMH* or *AMHR2* do not cause PMDS (*Picard et al., 2017*). Notably, parents of PMDS probands carrying heterozygote *AMH* or *AMHR2* mutations are fertile (*Picard et al., 2017*; *Josso et al., 2005*); however, detailed reproductive and olfactory phenotyping in these parents have not been reported. Furthermore, there are few studies examining the hormonal profile of patients with PMDS although spontaneous puberty is reported based on clinical observation (*Josso et al., 2005*). To assess pubertal and reproductive defects in PMDS patients or family members, detailed reproductive phenotyping will be necessary.

Taken together, these data demonstrate the pleiotropic roles of *AMH* and *AMHR2* in shaping internal genitalia and GnRH neuron migration. Different mechanistic actions of the mutants (i.e. recessive vs. dominant negative vs. haploinsufficiency) in combination with tissue-specific signaling pathway might guide the final phenotype.

*AMH* and *AMHR2* mutations affecting GN11 cell motility were found in both KS and nCHH individuals. We thus speculate that GnRH migratory defects could also occur in some cases of nCHH. This hypothesis challenges the current dogma, whereby defects in GnRH cell migration lead to KS and not nCHH (*Boehm et al., 2015*). Yet, increasing genetic evidence indicates that CHH genes do not segregate into 2 (i.e. KS and nCHH) but rather three categories: 1) KS only, 2) nCHH only and 3) both KS and nCHH. The latter includes genes involved in GnRH neuron migration such as *FGFR1*, *SEMA7A*, *AXL* (*Boehm et al., 2015*). We recently showed that GnRH neurons in human fetuses migrate into the brain in tight associations with vomeronasal and terminal nerves and not olfactory nerves (*Casoni et al., 2016*). Another recent study demonstrated that correct development of the OBs and axonal connection to the forebrain are not required for GnRH neuronal migration (*Taroc et al., 2017*), thus implying a stronger contribution of the vomeronasal/terminal nerve as a scaffold for the GnRH migration. Taken together, this evidence indicates that the vomeronasal and the terminal nerve play important roles in the ontogenesis and migration of GnRH neurons in vertebrates, and raises the intriguing possibility that some genetic forms of nCHH might be due to defective central projections of the vomeronasal/terminal nerves leading to subsequent alterations of the GnRH migratory process.

In conclusion, this work highlights the role of AMH/AMHR2 signaling in GnRH neuronal migration, hormone secretion and regulation of fertility, and identifies heterozygous mutations in *AMH* and *AMHR2* in CHH patients.

# Materials and methods

## Key resources table

| Reagent type (species) or resource | Designation | Source or reference | Identifiers | Additional information |
|---|---|---|---|---|
| Strain, strain background (*Mus musculus*) | C57BL/6J | Charles River | | |
| Strain, strain background (*M. musculus*) | Amhr2-Cre Knock-in | *Jamin et al., 2002* | | DOI: 10.1038/ng1003 |
| Strain, strain background (*M. musculus*) | Gnrh1 < GFP> | *Spergel et al., 1999* | | DOI: 10.1523/JNEUROSCI.19-06-02037.1999 |
| Recombinant DNA reagent | AMH-His | GeneCust | | Seq ref: NM_000479.3 |
| Recombinant DNA reagent | AMHR2-His | GeneCust | | Seq Ref: NM_000479.3 |
| Recombinant DNA reagent | AMH-p.Thr99Ser-His | This Paper | | |
| Recombinant DNA reagent | AMH-p.Pro151Ser-His | This Paper | | |
| Recombinant DNA reagent | AMH-p.Asp238Glu-His | This Paper | | |
| Recombinant DNA reagent | AMHR2-p.Gly445_Leu453del-His | This Paper | | |
| Cell line | GN11 | *Radovick et al., 1991* | Lab Stock | https://doi.org/10.1073/pnas.88.8.3402 GN11 cells were isolated from a male mouse |
| Cell line | GT1-7 | *Mellon et al., 1990* | Lab Stock; RRID:CVCL_0281 | https://doi.org/10.1016/0896-6273(90)90028-E GT1-7 cells were isolated from a mouse, unknown sex |
| Cell line | COS-7 | | Lab Stock; RRID:CVCL_0224 | COS-7 cells were isolated from a monkey |
| Transfected construct | Amhr2 SMARTpool siRNA | Dharmacon | #M-053605-00-0005 | |
| Transfected construct | Acvr1 SMARTpool siRNA | Dharmacon | #M-042047-01-0005 | |
| Transfected construct | Bmpr1a SMARTpool siRNA | Dharmacon | # M-040598-01-0005 | |
| Transfected construct | Bmpr1b SMARTpool siRNA | Dharmacon | # M-051071-00-0005 | |
| Transfected construct | Non-targeting siRNA control pool | Dharmacon | # D-001206-13-05 | |
| Antibody | Phospho-ERK1/2 (Thr202/Tyr204) (rabbit) | Cell Signaling | #9101L; RRID:AB_331646 | 1:1000 |
| Antibody | ERK1/2 (Thr202/Tyr204) (rabbit) | Cell Signaling | #9102L; RRID:AB_330744 | 1:1000 |
| Antibody | AMH (mouse) | Abcam | #Ab24542 ; RRID:AB_2801539 | 1:500 |
| Antibody | AMH (rabbit) | Abcam | #Ab103233; RRID:AB_10711946 | 1:500 |
| Antibody | AMHR2 (rabbit) | CASLO | Custom made #56G | 1:2000 |

*Continued on next page*

*Continued*

| Reagent type (species) or resource | Designation | Source or reference | Identifiers | Additional information |
|---|---|---|---|---|
| Antibody | GnRH (guinea pig) | Dr. Erik Hrabovszky, Institute of Experimental Medicine of the Hungarian Academy of Sciences, Budapest, Hungary | Lab Stock | 1:3000; https://doi.org/10.3389/fendo.2011.00080 |
| Antibody | Peripherin (Contactin1) (rabbit) | Millipore | #AB1530; RRID:AB_90725 | 1:1000 |
| Antibody | β-III tubulin (TUJ-1) (mouse) | Sigma Aldrich | #T8660; RRID:AB_477590 | 1:800 |
| Antibody | AMHR2 Neutralizing Antibody | R&D systems | #AF1618 ; RRID: AB_2226485 | 1:200 |
| Antibody | TAG-1 (goat) | R&D systems | AF2215 | |
| Antibody | Actin (mouse) | Sigma Aldrich | #A5316; RRID:AB_476743 | 1:1000 |
| Antibody | Donkey anti-rabbit IgG AlexaFluor 488 (H + L) | Molecular Probes | #A-21026; RRID:AB_141708 | 1:500 |
| Antibody | Donkey anti-rabbit IgG AlexaFluor 555 (H + L) | Molecular Probes | #A-31572; RRID:AB_162543 | 1:500 |
| Antibody | Donkey anti-mouse IgG AlexaFluor 488 (H + L) | Molecular Probes | #A-21202; RRID:AB_141607 | 1:500 |
| Antibody | Donkey anti-mouse IgG AlexaFluor 555 (H + L) | Molecular Probes | #A-31570; RRID:AB_2536180 | 1:500 |
| Antibody | Donkey anti-goat IgG AlexaFluor 488 (H + L) | Molecular Probes | #A-11055; RRID:AB_142672 | 1:500 |
| Antibody | Donkey anti-goat IgG AlexaFluor 555 (H + L) | Molecular Probes | #A-21432; RRID:AB_141788 | 1:500 |
| Antibody | Donkey anti-goat IgG AlexaFluor 647 (H + L) | Molecular Probes | #A-21447; RRID:AB_141844 | 1:500 |
| Antibody | Donkey anti-guinea pig IgG AlexaFluor 488 (H + L) | Jackson ImmunoResearch | #706-545-148; RRID:AB_2340472 | 1:500 |
| Antibody | Horse anti-mouse IgG peroxidase labelled | Vector | #PI-2000; RRID:AB_2336177 | 1:5000 |
| Sequence-based reagent | Amh Taqman gene expression assay | Thermofisher Scientific | Mm00431795_g1 | |
| Sequence-based reagent | GnRH Taqman gene expression assay | Thermofisher Scientific | Mm01315605 | |
| Sequence-based reagent | Amhr2 Taqman gene expression assay | Thermofisher Scientific | Mm00513847_m1 | |
| Sequence-based reagent | Acvr1 Taqman gene expression assay | Thermofisher Scientific | Mm01331069_m1 | |

*Continued on next page*

*Continued*

| Reagent type (species) or resource | Designation | Source or reference | Identifiers | Additional information |
|---|---|---|---|---|
| Sequence-based reagent | Bmpr1a Taqman gene expression assay | Thermofisher Scientific | Mm00477650_m1 | |
| Sequence-based reagent | Bmpr1b Taqman gene expression assay | Thermofisher Scientific | Mm03023971_m1 | |
| Sequence-based reagent | Rn18s Taqman gene expression assay | Thermofisher Scientific | Hs99999901-s1 | |
| Sequence-based reagent | Actb Taqman gene expression assay | Thermofisher Scientific | Mm00607939 | |
| Peptide, recombinant protein | Recombinant Human AMH-C Fragment (goat) | R&D systems | #1737 MS; RRID:AB_2273957 | |
| Commercial assay or kit | Papain Dissociation System | Worthington | #LK003150 | |
| Commercial assay or kit | Lipofectamine 2000 | ThermoFisher Scientific | #11668019 | |
| Commercial assay or kit | AMH Access Dxi chemiluminescent immunoassay | Beckman Coulter | #B13127 | |
| Commercial assay or kit | GnRH EIA kit | Phoenix Pharmaceuticals Inc | #EK-040-02CE | |
| Commercial assay or kit | Annexin V Apoptosis Detection Kit | Thermofisher Scientific | #88-8007-74 | |
| Commercial assay or kit | SureSelect All Exon capture V2 | Agilent Technologies | #5190–9493 | |
| Commercial assay or kit | Gentra Puregene Blood Kit | Qiagen | #158389 | |
| Chemical compound, drug | Flurogold Tracer | Sigma Aldrich | #39286 | 1:1500 |
| Chemical compound, drug | MAPK Kinase inhibitor | Calbiochem | #U0126 | 10 µM |
| Software, algorithm | FACSDiva | BD Biosciences | 8.1 | http://www.bdbiosciences.com/sg/instruments/software/downloads/ |
| Software, algorithm | SDS | Applied Biosystems | 2.4.1 | https://www.thermofisher.com/fr/fr/home/technical-resources/software-downloads/applied-biosystems-7900ht-fast-real-timespcr-system.html |
| Software, algorithm | Data Assist | Applied Biosystems | 3.0.1 | https://www.thermofisher.com/fr/fr/home/technical-resources/software-downloads/dataassist-software.html |
| Software, algorithm | ImageJ | NIH | 3.0.1 | https://imagej.net/Welcome |
| Software, algorithm | IMARIS | Bitplane | 9.1 | https://imaris.oxinst.com/ |
| Software, algorithm | Photoshop | Adobe | 4.0 | https://www.adobe.com/la/products/photoshop.html |

*Continued on next page*

*Continued*

| Reagent type (species) or resource | Designation | Source or reference | Identifiers | Additional information |
|---|---|---|---|---|
| Software, algorithm | Illustrator | Adobe | 4.0 | https://www.adobe.com/products/illustrator.html |
| Software, algorithm | Prism 6 | Graphpad Software | 6.0 | https://www.graphpad.com/scientific-software/prism/ |
| Software, algorithm | Inspector Pro | La Vision Biotec | 4.0 | |
| Software, algorithm | Burrows-Wheeler Alignment Algorithm | | | http://bio-bwa.sourceforge.net/ |
| Software, algorithm | SnpEff | Switch Laboratoty | 4.0 | http://snpeff.sourceforge.net/ |
| Software, algorithm | dbNSFP | *Liu et al., 2011* | 2.9 | http://varianttools.sourceforge.net/Annotation/dbNSFP |
| Software, algorithm | Modeller | *Webb and Sali, 2016* | 9.2 | https://salilab.org/modeller/ |

## Animals and cell lines

C57BL/6J mice (Charles River, USA) were housed under specific pathogen-free conditions in a temperature-controlled room (21–22°C) with a 12 hr light/dark cycle and ad libitum access to food and water. *Amhr2*-Cre knock-in mice have been previously characterized (*Jamin et al., 2002*). *Gnrh1<GFP>* (*Spergel et al., 1999*) mice were a generous gift of Dr. Daniel J. Spergel (Section of Endocrinology, Department of Medicine, University of Chicago, IL). Mice were genotyped by PCR using primers listed in *Supplementary file 1*.

Animal studies were approved by the Institutional Ethics Committees of Care and Use of Experimental Animals of the Universities of Lille 2 (France). All experiments were performed in accordance with the guidelines for animal use specified by the European Union Council Directive of September 22, 2010 (2010/63/EU) and the approved protocol (APAFIS#13387–2017122712209790 v9) by the Ethical Committee of the French Ministry of Education and Research.

All efforts were made to minimize animal suffering and animal care was supervised by veterinarians and animal technicians skilled in rodent healthcare and housing. Mice of the appropriate genotype were randomly allocated to experimental groups.

COS-7 cells (originally from the lab stock of Luca Tamagnone lab), GN11 and GT1-7 cells (*Radovick et al., 1991*; *Mellon et al., 1990*) (originally from the lab stock of Pamela Mellon lab and from the lab stock of Sally *Radovick* lab) were grown in DMEM with 10% fetal bovine serum (Invitrogen). They were authenticated based on morphology, and DNA staining revealed no mycoplasma contamination.

## Immunofluorescence

Embryos were harvested at embryonic day E14.5 from black C57BL/6 mice. Heads from the embryos were washed thoroughly in cold 0.01M PBS, fixed in fixative solution [4% paraformaldehyde (PFA), 0.01M PBS, pH 7.4] for 6–8 hr at 4°C and cryoprotected in 30% sucrose overnight at 4°C. The following day, heads were embedded in OCT embedding medium (Tissue-Tek, Torrence, CA), frozen on dry ice, and stored at −80°C until sectioning. The embryo heads were coronally cryosectioned (Leica Microsystems, Wetzlar Germany) at 16 μm intervals directly onto slides and stored at −80°C until use. Adult mice were anesthetized with 80 mg/kg of ketamine-HCl and 8 mg/kg xylazine-HCl and transcardially perfused with 40 ml of saline, followed by 100 ml of 4% PFA, pH 7.4. Brains were collected, postfixed in the same fixative for 2 hr at 4°C and cryoprotected in 30% sucrose overnight. Embedded in OCT embedding medium, frozen on dry ice and stored at −80°C until cryosectioning. Adult brains were sectioned at 35 μm using the cryostat and stored in anti-freeze medium at −20°C until use. Immunolabelling for mouse and human samples was completed as follows: sections were thawed at RT before 3 × 5 min washes in 0.01M PBS. Sections were then incubated with primary

antibodies (Key Resources Table) in a solution containing 10% normal donkey serum and 0.3% Triton X100 for 3 days at 4°C. 3 × 5 min washes in 0.01M PBS were followed by incubation in appropriately conjugated secondary antibodies (Key Resources Table) for 1 hr before incubation with Hoechst 1:1000. After 3 × 5 min washes in 0.01M PBS sections were coverslipped using Mowiol as an anti-fade mounting medium.

## Nasal explants

Embryos were obtained from timed-pregnant animals. Vaginal plug dates were designed as E0.5. Nasal pits of E11.5 WT C57BL/6J mice were isolated under aseptic conditions in Grey's Balanced Salt Solution (Invitrogen) enriched with glucose (Sigma-Aldrich) and maintained at 4°C until plating. Explants were placed onto glass coverslips coated with 10 µl of chicken plasma (Cocalico Biologicals, Inc). Thrombin (10 µl; Sigma-Aldrich) was then added to adhere (thrombin/plasma clot) the explant to the coverslip. Explants were maintained in defined serum-free medium (SFM) (*Fueshko and Wray, 1994*) containing 2.5 mg/ml Fungizone (Sigma-Aldrich) at 37°C with 5% CO (*Wray et al., 1989*) for up to 30 days in vitro (div). From culture days 3 to 6, fresh medium containing fluorodeoxy-uridine ($8 \times 10^{-5}$ M; Sigma-Aldrich) was provided to inhibit the proliferation of dividing olfactory neurons and non-neuronal explant tissue. The medium was replaced with fresh SFM twice a week.

## In utero injections

Vaginal plug dates were designed as E0.5. Timed-pregnant mice (*n* = 2) carrying E12.5 embryos were anesthetized with isoflurane, and the uterine horns were gently placed outside the abdominal cavity and constantly hydrated with 35°C sterile saline. Using a Nanofil syringe and a 35 GA needle attachment (World Precision Instruments), 2 µl containing 0.4 µg of Amhr2 Neutralizing Antibody (Amhr2-NA, 1:200, R and D system, AF1618) and Fluorogold tracer 1:1500 (Sigma Aldrich, #39286) diluted in saline was injected intra-utero in the olfactory placode of each embryo of one uterine horn. In order to consistently obtain control and Amhr2-NA treated embryos from the same pregnant animals and limit biases associated with the staging of the embryonic development, the embryos of the contralateral horn were injected with saline and Fluorogold of both dams. The concentration of AMHR2-NA was determined based on the manufacturer's recommendations. The uteri were gently returned and the mothers sutured and monitored for few days. Embryos were collected at embryonic day 14.5 (E14.5), fixed, cryoprotected, frozen and cut as described above (Immunofluorescence). Fluorogold was used in order to verify the specificity of the injection sites and only embryos confirmed as optimal hits (fluorogold fluorescence within the olfactory epithelium) were used for the GnRH quantitative analysis (*n* = 4 per treatment group, from two independent dams).

## GnRH cell counting

Serial sagittal sections (16 µm) from E14.5 WT embryos, (*n* = 4 per group) were prepared as described above. Quantitative analysis of GnRH neuronal number, as a function of location, was performed over four regions (the nasal compartment, the nasal/forebrain junction, ventral forebrain and cortex). Serial coronal sections (35 µm) of $Amhr2^{+/+}$, $Amhr2^{+/-}$ and $Amhr2^{-/-}$ mouse brains were used to count the total number of GnRH cells throughout the entire brain and combined to give group means ± SEM. Vaginal plug dates were designed as E0.5.

## Fluorescence activated cell sorting (FACS)

Embryos were harvested at E12.5 from timed-pregnant *Gnrh1 <GFP>* mice, previously anesthetized with an intraperitoneal injection of 80 mg/kg of ketamine-HCl and 8 mg/kg xylazine-HCl and sacrificed by cervical dislocation. Juvenile (P12) and adult female mice (3 months old) were anesthetized with 80 mg/kg of ketamine-HCl and 8 mg/kg xylazine-HCl before being sacrificed by cervical dislocation. Microdissections from embryonic nasal region and post-natal/adult hypothalamic preoptic region were enzymatically dissociated using Papain Dissociation System (Worthington, Lakewood, NJ) to obtain single-cell suspensions as previously described (*Messina et al., 2016*). After dissociation, the cells were physically purified using a FACSAria III (Beckman Coulter) flow cytometer equipped with FACSDiva software (BD Biosciences). The sort decision was based on measurements of GFP fluorescence (excitation: 488 nm, 50 mW; detection: GFP bandpass 530/30 nm, autofluorescence bandpass 695/40 nm) by comparing cell suspensions from GnRH-GFP and wild-type animals.

For each animal, 500 GFP-positive cells were sorted directly into 8 µl extraction buffer: 0.1% Triton X-100 (Sigma-Aldrich) and 0.4 U/µl RNaseOUT (Life Technologies). Captured cells were used to synthesize first-strand cDNA using the protocol detailed below.

## Immortalized cell cultures

GN11, GT1-7 and COS-7 cells were grown in monolayers at 37°C under 5% $CO_2$, in DMEM (Thermo-Fisher, Invitrogen) containing 1 mM pyruvate, 2 mM L-glutamine (ThermoFisher, Invitrogen), 100 µg/ml streptomycin, 100 U/ml penicillin and 9 mg/ml glucose (MP Biomedicals, Santa Ana, CA), supplemented with 10% fetal bovine serum (complete medium). Cells were maintained below full confluence by trypsination and seeding onto 10 cm$^2$ dishes. Cells used for experiments were between their third and eighth passage. Cells were treated with recombinant human AMH (1737-MS; R&D systems) at the concentrations ranging from 10 ng/ml to 250 ng/ml.

## Quantitative RT-PCR

For gene expression analyses, cDNA obtained from RT-PCR were reverse transcribed using Super-Script III Reverse Transcriptase (ThermoFisher, Invitrogen). Real-time PCR was carried out on Applied Biosystems 7900HT Fast Real-Time PCR System using exon-span-specific TaqMan Gene Expression Assays (Applied Biosystems, Carlsbad CA). The list of primers used for these experiments is the following: *Amh* (Mm00431795_g1), *Gnrh1* (Mm01315605), *Amhr2* (Mm00513847_m1); *Acvr1* (Mm01331069_m1); *Bmpr1a* (Mm00477650_m1); *Bmpr1b* (Mm03023971_m1). Control housekeeping genes: *Rn18s* (Hs99999901-s1) and *Actb* (Mm00607939). Amperase activation was achieved by heating to 50°C for 2 min, before denaturing at 95°C for 20 s, followed by 40 cycles of 1 s 95°C with a 20 s extension time at 60°C. Gene expression data were analyzed using SDS 2.4.1 and Data Assist 3.0.1 software (Applied Biosystems, Carlsbad, CA), with *ActB* and *Rn18s* as control house-keeping mRNA following a standardized procedure (*Schmittgen and Livak, 2008*). Values are normalized relative to control values and expressed, as appropriate, to 1.

## Western blot

Culture plates were frozen as described above quickly thawed and protein immediately extracted with 150 µl of freshly prepared lysis buffer [25 mM Tris pH 7.4, 50 mM β-glycerophosphate, 1% Triton x100, 1.5 mM EGTA, 0.5 mM EDTA, 1 mM sodium orthovanadate, 10 µg/ml Leupeptin and Pepstatin A, 10 µg/ml aprotinin, 100 µg/ml PMSF (reagents sourced from Sigma Aldrich, St. Louis, MO)]. Protein extracts were then homogenized using a 26 gauge needle before centrifuging at 12.000 g for 15 mins at 4°C. The supernatant was recovered and protein quantified using the Bradford method (BioRad, Hercules, CA). 1x sample and 4x loading buffer (ThermoFisher, Invitrogen) were added to the samples, which were then boiled for 10 min before electrophoresis at 120V for 100 mins in 4–12% tris-acetate precast SDS-polyacrylamide gels according to the protocol supplied with the NuPAGE system (ThermoFisher, Invitrogen). After size fractionation, the proteins were transferred onto a PVDF membrane (0.2 µm pore size, LC2002; Invitrogen, Carlsbad, CA) in the blot module of the NuPAGE system (ThermoFisher, Invitrogen) maintained at 1A for 75 min at room temperature (RT). Blots were blocked for 1 hr in tris-buffered saline with 0.05% Tween 20 (TBST) and 5% non-fat milk at RT, incubated overnight at 4°C with their respective primary antibody in TBST 5% bovine serum albumin (Sigma Aldrich, Cat A7906), and washed four times with TBST before being exposed to horseradish peroxidase-conjugated secondary antibodies diluted in 5% non-fat milk TBST for 1 hr at RT. The immunoreactions were detected with enhanced chemiluminescence (NEL101; PerkinElmer, Boston, MA).

## Cell transfections for functional validations

Expression vectors encoding for human AMH and AMHR2 were obtained from Genescript. Briefly, a cDNA containing the entire coding region of the human *AMH* transcript (NM_000479.3) and *AMHR2* (NM_020547.3) were inserted into a modified pcDNA3.1+expression vector containing a his-tag at the 5'end (GeneCust). The plasmids encoding the variants (AMH: p.Thr99Ser, p.Pro151Ser, p. Asp238Glu, and AMHR2: p.Gly455_Leu453del) were generated by site directed mutagenesis using the QuickChange XLII Kit (Stratagene) and confirmed by Sanger sequencing. Primers flanking the mutations (see primers list in *supplementary file 1*) were used for subsequent PCR amplifications.

COS-7, GN11 or GT1-7 cells were grown to 70% confluence in 10 cm$^2$ dish without the presence of antibiotics in preparation for transfection. For each plasmid, oligomer-Lipofectamine 2000 complexes were prepared as follows: vectors were diluted in 500 µl OptiMEM Reduced Serum Medium without serum and gently mixed, for a final concentration of 400 nM. Lipofectamine 2000 (Thermo-Fisher, Invitrogen) was mixed gently before use, then diluted 10 µl in 500 µl OptiMEM (Thermo-Fisher, Invitrogen). Tubes were gently mixed and incubated for 5 min at room temperature. After the 5-min incubation, the diluted vector was combined with the diluted Lipofectamine 2000 and incubated for 20 min at room temperature. During the incubation, cells were trypsinized and dissociated, then re-suspended in the lipofectamine containing vector mixture. Cells were then incubated at 37°C in a 5% $CO_2$ incubator for 48 hr, changing the medium to OptiMEM supplemented with 5% fetal bovine serum after 6 hr. Conditioned media was collected for AMH or GnRH quantitation and transfected cells used for either western blotting or transwell migration assays.

## AMH quantification

AMH levels in conditioned media were measured by an automatic chemoluminescent immunoassay on a Dxi system (Beckman Coulter, France) after a 1/10 dilution in the Sample Diluent A. This assay detects proAMH and the cleaved $AMH_{N,C}$ complex. The limit of quantification of the assay is 0.57 pmol/L with an intra- and inter-assay imprecision less than 5%.

## Determination of GnRH secretion

GT1-7 cells were transiently transfected in OptiMem with either *AMH* WT or p.Thr99Ser h*AMH* variants. 48 hr later, the medium was collected and frozen until EIA measurement. In another set of experiments, GT1-7 cells were transiently transfected with either *AMHR2* WT or with the *AMHR2* CHH variants. 48 hr later, the cells were treated for 4 hr with either PBS or recombinant AMH (1737-MS; R&D systems, 50 ng/ml). Finally, the medium was frozen until EIA measurement. The collected media from these experiments were analyzed for GnRH content following a GnRH EIA protocol (EK-040-02CE, Phoenix Pharmaceuticals Inc, CA).

## Transwell migration assay

Transwell chambers were used according to manufacturer's instructions (Falcon). In brief, GN11 cells grown in complete medium until sub-confluence were harvested and re-suspended at a density of 1 × 10$^5$ cells/µl in SFM. Cells were seeded on the upper side of 8 µm pore membranes and incubated for 12 hr with SFM, human recombinant AMH (1737-MS; R&D systems, 50–250 ng/ml) or with DMEM supplemented with 10% fetal bovine serum. Each factor (serum, AMH, inhibitors and antibodies) was placed on the upper and lower chamber of the transwell. GN11 cells were incubated in the presence of recombinant AMH (50 ng/ml) together with MAPK inhibitor (UO126; Calbiochem) at a concentration of 10 µM, as previously described (*Balland et al., 2014*). In another set of experiments, GN11 cells were treated for 12 hr with Amhr2-NA (R and D system, AF1618), at the same concentration (1:200) used for the *in utero* injections experiments (*Figure 2*), in the presence or absence of recombinant AMH (50 ng/ml). Cells on the upper side of the filters were mechanically removed and cells on the lower side fixed in 4% PFA for 30 min before nuclei labelling with DAPI. Four non-overlapping regions were imaged per membrane using a Zeiss 20x objective (N.A. 0.8) mounted on a Axio Imager Z2 light microscope (Zeiss), with nuclei counted using an ImageJ plugin (National Institute of Health, Bethseda) and averaged to produce an average per well. n for each experiment is detailed in the figure legends.

## siRNA transfections

GN11 cells were grown to 70% confluence in 10 cm$^2$ dish without the presence of antibiotics in preparation for transfection. For siRNA experiments, GN11 cells were transiently transfected with 75 nM SMARTpool siRNA targeting mouse *Amhr2*, *Acvr1*, *Bmpr1a*, *Bmpr1b*, or 75 nM nontargeting SMARTpool siRNA (siRNA NT) as negative control (Dharmacon, Horizon Discovery LTD, Cambridge, UK). Gene knockdown was assessed by quantitative PCR.

## Ovarian histology

Ovaries were collected from 6-month-old mice, immersion-fixed in 4% PFA solution and stored at 4° C. Paraffin-embedded ovaries were sectioned at a thickness of 5 μm (histology facility, University of Lille 2, France) and stained with hematoxylin-eosin (Sigma Aldrich, Cat # GHS132, HT1103128). Sections were examined throughout the ovary. Corpora lutea (CL) were classified and quantified as previously reported (*Caldwell et al., 2017*). To avoid repetitive counting, CL were counted every 100 μm by comparing the section with the preceding and following sections. CL were characterized by a still present central cavity, filled with blood and follicular fluid remnants or by prominent polyhedral to round luteal cells.

## Pulsatile LH measurement

Mice were habituated with daily handling for 3–4 weeks. Blood samples (5 μl) were taken from the tail in 10 min intervals for 2 hr (between 12h00 hours and 15h00 hours), diluted in PBS-Tween and immediately frozen. LH levels were determined by sandwich ELISA (*Steyn et al., 2013*). A 96-well high-affinity binding microplate (9018; Corning) was coated with 50 μl of capture antibody (monoclonal antibody, anti-bovine LH beta subunit, 518B7; University of California) at a final dilution of 1:1000 (in 1 x PBS, 1.09 g of Na2HPO4 (anhydrous), 0.32 g of NaH2PO4 (anhydrous) and 9 g of NaCl in 1000 ml of distilled water) and incubated overnight at 4° C. Wells were incubated with 200 ml of blocking buffer (5% (w/v) skim milk powder in 1 x PBS-T (1 x PBS with 0.05% Tween 20) for 2 hr at room temperature. A standard curve was generated using a twofold serial dilution of mLH (reference preparation, AFP-5306A; National Institute of Diabetes and Digestive and Kidney Diseases - National Hormone and Pituitary Program (NIDDK-NHPP)) in 0.2% (w/v) bovine serum albumin 1 x PBS-T. The LH standards and blood samples were incubated with 50 ml of detection antibody (polyclonal antibody, rabbit LH antiserum, AFP240580Rb; NIDDK-NHPP) at a final dilution of 1:10,000 for 1.5 hr (at RT). Each well containing bound substrate was incubated with 50 μl of horseradish peroxidase-conjugated antibody (polyclonal goat anti-rabbit; Vector) at a final dilution of 1:10,000. After a 1.5 hr incubation, 100 μl of o-phenylenediamine (002003; Invitrogen), substrate containing 0.1% $H_2O_2$ was added to each well and left at RT for 30 min. The reaction was stopped by the addition of 50 ml of 3M HCl to each well, and absorbance of each well was read at a wavelength of 490 nm. Pulses were confirmed using DynPeak (*Vidal et al., 2012*).

## Fertility test

The reproductive competency of these animals was determined by pairing the following mice: $Amhr2^{+/+}$ males mated with $Amhr2^{+/+}$ females, $Amhr2^{+/-}$ females, or with $Amhr2^{-/-}$ females, or inversely, for a period of 3 months. One male and one female were housed in each cage during the constant breeding protocol. Each litter was sacrificed at birth to allow the dams to re-enter estrous cyclicity within a few days. Number of pups/litter, fertility index (number of litters per female per month, averaged during the 3 months), and time to first litter (number of days to first litter after pairing) were quantified per pairing.

## iDISCO

Experiments were performed as previously described (*Renier et al., 2014*) and detailed below.

### Sample pre-treatment with methanol

Samples were washed in PBS (twice for 1 hr), followed by 50% methanol in PBS (once for 1 hr), 80% methanol (once for 1 hr) and 100% methanol (twice for 1 hr). Next, samples were bleached in 5% H2O2 in 20% DMSO/methanol (2 ml 30% $H_2O_2$/2 ml DMSO/8 ml methanol, ice cold) at 4°C overnight. Next, samples were washed in methanol (twice for 1 hr), in 20% DMSO/methanol (twice for 1 hr), 80% methanol (once for 1 hr), 50% methanol (once for 1 hr), PBS (twice for 1 hr), and finally, PBS/0.2% TritonX-100 (twice for 1 hr) before proceeding to the staining procedures.

### Whole-mount immunostaining

Samples were incubated at 37°C on an adjustable rotator in 10 ml of a blocking solution (PBSGNaT) of 1X PBS containing 0.2% gelatin (Sigma), 0.5% Triton X-100 (Sigma-Aldrich) and 0.01% NaAzide for three nights. Samples were transferred to 10 ml of PBSGNaT containing primary antibodies (Key

resources table) and placed at 37°C in rotation for 7 days This was followed by six washes of 30 min in PBSGT at RT and a final wash in PBSGT overnight at 4°C. Next, samples were incubated in secondary antibodies (1:400, Alexa 568, Alexa 647) diluted in 10 ml PBSGNaT for 2 days at 37°C in a rotating tube. After six 30 min washes in PBS at room temperature, the samples were stored in PBS at 4°C in the dark until clearing.

## Tissue clearing

All incubation steps were performed at RT in a fume hood, on a tube rotator at 14 rpm covered with aluminium foil to avoid contact with light. Samples were dehydrated in a graded series (50%, 80%, and 100%) of tetrahydrofurane (THF; anhydrous, containing 250 ppm butylated hydroxytoluene inhibitor, Sigma-Aldrich) diluted in $H_2O$ as follow: 1) 50% THF overnight at RT; 2) 80% THF 1 hr at RT; 3) 100% THF 1h30 at RT; 4) 100% THF 1h30 at RT. This was followed by a delipidation step of 30–40 min in 100% dichloromethane (DCM; Sigma-Aldrich). Samples were cleared in dibenzylether (DBE; Sigma-Aldrich) for 2 hr at RT on constant agitation and in the dark. Finally, samples were moved into fresh DBE and stored in glass tube in the dark and at RT until imaging. We could image samples, as described below, without any significant fluorescence loss for up to 6 months.

## Imaging

3D imaging was performed as previously described (*Belle et al., 2014*). An ultramicroscope (LaVision BioTec) using ImspectorPro software (LaVision BioTec) was used to perform imaging. The light sheet was generated by a laser (wavelength 488 or 561 nm, Coherent Sapphire Laser, LaVision BioTec) and two cylindrical lenses. A binocular stereomicroscope (MXV10, Olympus) with a 2 × objective (MVPLAPO, Olympus) was used at different magnifications (1.6×, 4×, 5×, and 6.3×). Samples were placed in an imaging reservoir made of 100% quartz (LaVision BioTec) filled with DBE and illuminated from the side by the laser light. A PCO Edge SCMOS CCD camera (2560 × 2160 pixel size, LaVision BioTec) was used to acquire images. The step size between each image was fixed at 2 μm.

## Image analysis

For confocal observations and analyses, an inverted laser scanning Axio observer microscope (LSM 710, Zeiss, Oberkochen, Germany) with EC Plan NeoFluor 10×/0.3 NA, 20×/0.5 NA and 40×/1.3 NA (Zeiss, Oberkochen, Germany) objectives were used (Imaging Core Facility of IFR114 of the University of Lille, France).

Images, 3D volume, and movies were generated using Imaris x64 software (version 7.6.1, Bitplane). Stack images were first converted to imaris file (.ims) using ImarisFileConverter and 3D reconstruction was performed using 'volume rendering'. Optical slices of samples were obtained using the 'orthoslicer' tools. The surface of the samples was created using the 'surface' tool by creating a mask around each volume. 3D pictures were generated using the 'snapshot' tool. ImageJ (National Institute of Health, Bethesda) and Photoshop CS6 (Adobe Systems, San Jose, CA) were used to process, adjust and merge the photomontages. Figures were prepared using Adobe Photoshop and Abode Illustrator CS6.

## Human CHH subjects

The CHH cohort included 180 probands (105 KS and 75 nCHH). The majority of the patients were male (n = 127). The diagnosis of CHH was made on the basis of: i) absent or incomplete puberty by 17 years of age; ii) low/normal gonadotropin levels in the setting of low serum testosterone/estradiol levels; and iii) otherwise normal anterior pituitary function and normal imaging of the hypothalamic-pituitary area (*Pitteloud et al., 2002*). Olfaction was assessed by self-report and/or formal testing (*Lewkowitz-Shpuntoff et al., 2012*). When available, family members were included for genetic studies. This study was approved by the ethics committee of the University of Lausanne, and all participants provided written informed consent prior to study participation.

## Human case summaries

### Family # 1, Patient II-1
*AMH* p.Thr99Ser (heterozygous)

The caucasian male proband consulted our clinic at age 32 for symptomatic hypogonadism accompanied by mild anemia and oligospermia. He was previously diagnosed with delayed puberty at age 17 but was never offered testosterone replacement nor was he followed up to ensure pubertal completion. Physical examination revealed partial but incomplete puberty (testicular volume 12 ml bilaterally, pubic hair Tanner IV) as well as eunuchoid proportions (arm span 184 cm for height of 176 cm). Targeted clinical exam detected slight hyperlaxity, high-arched palate and pectum excavatum. Blood tests confirmed a hypogonadotropic hypogonadism (testosterone 7.5 nmol/l, LH 2.7 U/l, FSH 3.6 U/l) without dysfunction of other pituitary axes. Formal olfactory testing confirmed normal sense of smell. Pituitary MRI and GnRH-stimulation test were normal. Bone density scan typically detected osteopenia at the lumbar spine. Family history included delayed puberty and growth in his father while mother's puberty was normal (menarche at 11 years old). The proband harbors a heterozygous mutation in *AMH* inherited by his father who as stated above has a partial phenotype (delayed puberty). He also harbors a *GNRHR* variant, which was not considered as pathogenic or likely pathogenic by our filtering process, due to its presence only at heterozygous state for a gene with autosomal recessive transmission mode. Given his request for fertility, HCG treatment was introduced, allowing for increase of testosterone and stimulation of spermatogenesis.

### Family # 2, Patient II-1
*AMH* p.Pro151Ser (heterozygous)

This anosmic male proband was diagnosed at 2 years old for unilateral cryptorchidism, treated by left orchidopexy. Subsequent follow up showed no signs of puberty at 17.5 years with prepubertal testes (volume 2 ml bilaterally). Serum testosterone was low while gonadotropins were undetectable (LH/FSH < 1.0 U/L). Formal smell testing confirmed anosmia (UPSIT: 11/40, <5th %ile) and he was diagnosed with Kallmann syndrome. Intramuscular injections of testosterone were initiated with good effect on virilization. A cranial MRI showed normal pituitary size and absent olfactory bulbs. Family history was negative for pubertal delay. Parents refused to participate in the genetic study and undergo smell testing. The patient harbors a rare mutation in *AMH* with no changes in known CHH genes. A pelvic MRI was performed in July 2017 and showed no argument in favor of PMDS.

### Family # 3, Patient II-2
*AMG* p.Asp238Gln (heterozygous)

The female proband originated from Kazakhstan consulted us at age of 27.9 years for infertility. She presented with primary amenorrhea at age 17. She described onset of breast development at 13 years, which rapidly stalled (Tanner II-III). She was placed on estrogen-progesterone replacement (estradiol/dihydrogesterone) at age 18. She remained amenorrheic during multiple treatment pauses. When assessed in our clinic and several months after withdrawal of estrogen pills, hypogonadotropic hypogonadism was confirmed (estradiol 0.05 nmol/l, LH 0.5 U/l, FSH 1.4 U/l) with otherwise normal pituitary function and adequate response to GnRH stimulation (LH baseline 0.4 U/l, peak 6.3 U/l; FSH baseline 1.2 U/l, peak 6.4 U/l). A 10 hr frequent sampling did not detect any LH pulses. Olfactory assessment by 12-item Sniffin' Sticks revealed hyposmia (12/16). Her physical status was notable for eunuchoid proportions and moderate scoliosis without dysmorphic features. Cranial MRI showed a small pituitary gland, while bone density scan indicated osteoporosis. Polycystic ovaries syndrome was excluded as well as a non-classic congenital adrenal hyperplasia. We concluded to KS diagnosis with partial GnRH deficiency and resumed estrogen/dihydrogesterone treatment. Family history included delayed puberty in the father (first shaving at age 18, continued growing until age 24). The latter was also anosmic, while the mother exhibited normal puberty (menarche at 13 years old) and olfactive function. The patient harbors an *AMH* mutation, inherited by the partially affected father. No changes in known CHH genes were seen.

## Family # 4, Patient II-1
### AMHR2 p.Gly445_Leu453del (heterozygous)

The female Caucasian patient presented with primary amenorrhea and absent pubertal development at age 17. She remained amenorrheic until age 22, and then was offered oral contraceptives (estradiol, norgestrel). Endocrinology assessment in her native country (Serbia) led to diagnosis of hypogonadotropic hypogonadism at age 33 (estradiol <0.04 nmol/l, LH <2.0 U/l, FSH 0.1 U/l) with otherwise normal pituitary function, assessed by an insulin tolerance test. Similarly, LHRH stimulation test showed adequate pituitary response (LH baseline 2.0 U/l, peak 6.5 U/l; FSH baseline 1.8 U/l, peak 6.3 U/l). Pituitary MRI was normal as well as formal smell test (Sniffin' Sticks 14/16, > 25th %ile). The patient consulted us to discuss ovulation induction by pulsatile GnRH treatment. After withdrawal of estrogen pills, hypogonadotropic hypogonadism was confirmed (estradiol <0.04 nmol/l, LH 0.4 U/l, FSH 1.4 U/l). Physical exam did not show any associated phenotypes. Family history was unremarkable for pubertal timing but her mother exhibited history of cleft lip/palate, corrected surgically at infancy. Detailed history of the father was impossible as he was deceased. A half paternal brother had normal puberty and fathered a child without difficulty. The patient harbored a heterozygous deletion in *AMHR2*. Sequencing of *AMHR2* in the patient's mother and half paternal brother showed no mutation.

## Genetic studies

Genomic DNA was extracted from peripheral-blood samples using the Puregene Blood Kit (Qiagen), following the manufacturer's protocol. Exome capture was performed using the SureSelect All Exon capture v2 and v5 (Agilent Technologies, Santa Clara, CA) and sequenced on the HiSeq2500 (Illumina, San Diego, CA) at BGI (BGI, Shenzen, PRC). Raw sequences (fastq files) were analyzed using an in-house pipeline that utilizes the Burrows-Wheeler Alignment algorithm (BWA) (*Li and Durbin, 2009*) for mapping the reads to the human reference sequence (GRCh37), and the Genome Analysis Toolkit (GATK) (*DePristo et al., 2011*) for the detection of single nucleotide variants (SNVs) and insertion/deletions (Indels). The resulting variants were annotated using SnpEff version 4.0 (*Cingolani et al., 2012*) and dbNSFP version 2.9 (*Liu et al., 2013*) to calculate minor allele frequency (MAF).

We evaluated coding exons and intronic splice regions (≤6 bp from the exons) of the known CHH genes for pathogenic and likely pathogenic variants according to ACMG guidelines (*Richards et al., 2015*). The included CHH genes are: *ANOS1* (NM_000216.2), *SEMA3A* (NM_006080), *FGF8* (NM_033163.3), *FGF17* (NM_003867.2), *SOX10* (NM_006941), *IL17RD* (NM_017563.3), *AXL* (NM_021913), *FGFR1* (NM_023110.2), *HS6ST1* (NM_004807.2), *PCSK1* (NM_000439), *LEP* (NM_000230), *LEPR* (NM_002303), *FEZF1* (NM_001024613), *NSMF* (NM_001130969.1), *PROKR2* (NM_144773.2), *WDR11* (NM_018117), *PROK2* (NM_001126128.13), *GNRH1* (NM_000825.3), *GNRHR* (NM_000406.2), *KISS1* (NM_002256.3), *KISS1R* (NM_032551.4), *TAC3* (NM_013251.3), and *TACR3* (NM_001059.2).

Forty-four probands harbored pathogenic or likely pathogenic variants in the known CHH genes, and were excluded for subsequent analysis. The remaining 136 probands were then evaluated for mutations in *AMH* and *AMHR2*. Only variants with MAF <0.1% were used for subsequent analysis. *AMH* and *AMHR2* variants were confirmed by Sanger sequencing on both strands with duplicate PCR reactions and are described according to HGVS nomenclature (*den Dunnen and Antonarakis, 2000*).

## Computational modelling of the p.Gly445_Leu453del *AMHR2* intracellular domain

For the WT AMHR2 kinase domain, a previous model (referred as WT) based on the crystallographic structure of the human activin type II B receptor (PDB code 2qlu), which shares ~35% sequence identity with AMHR2, was used (for more details see *Belville et al., 2009*). The 3D-model bearing the p.Gly445_Leu453del deletion (referred as DEL) was generated on the basis of the WT model, using the Modeler V9.2 package (*Sali and Blundell, 1993*) and evaluated by Errat (*Colovos and Yeates, 1993*). The three-dimensional models generated were submitted to a 100 ns molecular dynamics simulation. Both systems were set up following the same protocol and using the xleap program. The system was embedded in a cubic box with edges at 15 Å from the protein and sodium ions were

added to neutralize the simulation cell (4 and 6 for WT and DEL models, respectively). The protein was described by the ff14sb forcefield (*Maier et al., 2015*) water molecules with the TIP3P (*Jorgensen et al., 1983*) one and the ions94.lib library was used for the sodium cations. Non-bonded interactions were calculated with a cutoff of 10 Å, whereas long range electrostatic interactions were calculated with the Ewald Particle Mesh method (*Essmann et al., 1995*). A time step of 1 fs was used to integrate the equation of motion with a Langevin integrator (*Schneider and Stoll, 1978*; *Brünger et al., 1984*). Constant temperature and pressure were achieved by coupling the systems to a Monte Carlo barostat at 1.01325 bar. Bonds involving hydrogen atoms were constrained using the SHAKE algorithm (*Ryckaert et al., 1977*). The simulations were performed with OpenMM 7.0 (*Eastman and Pande, 2015*) following a standard protocol included into OMMProtocol application (https://github.com/insilichem/ommprotocol): model systems were initially energy minimized (3000 steps); then, thermalization of water molecules and side chains was achieved by increasing the temperature from 100 K up to 300 K; finally, 100 ns of production simulations were performed. For the model containing the deletion, simulations were performed in triplicate. Molecular graphics were produced with the UCSF Chimera package (*Pettersen et al., 2004*), except for the RMSF one that was done with VMD (*Humphrey et al., 1996*).

## Collection and processing of human fetuses

Tissues were made available in accordance with French bylaws (Good practice concerning the conservation, transformation and transportation of human tissue to be used therapeutically, published on December 29, 1998). The studies on human fetal tissue were approved by the French agency for biomedical research (Agence de la Biomédecine, Saint-Denis la Plaine, France, protocol n°: PFS16–002). Non-pathological human fetuses (11 weeks post-amenorrhea, *n* = 2, females) were obtained from voluntarily terminated pregnancies after obtaining written informed consent from the parents (Gynaecology Department, Jeanne de Flandre Hospital, Lille, France). Fetuses were fixed by immersion in 4% paraformaldehyde (PFA) at 4°C for 7 days. The tissues were then cryoprotected in 30% sucrose/PBS at 4°C overnight, embedded in Tissue-Tek OCT compound (Sakura Finetek, USA), frozen in dry ice and stored at −80°C until sectioning. Frozen samples were cut serially at 20 µm using a Leica CM 3050S cryostat (Leica Biosystems Nussloch GmbH, Germany) and immunolabeled as described above and as previously described (*Casoni et al., 2016*).

## Sex determination of human fetuses

Sex determination of two human fetuses (Gestational weeks 11: GW11) was obtained by isolating DNA from extracted tissues using the NucleoSpin Tissue Kit (Macherey-Nagel), according to manufacturer instructions, and the extracted DNA was stored at −20°C until use. The DNA concentration (absorbance at 260 nm) and purity (A260/A280 ratio) were assessed using the NanoDrop 1000 Spectrophotometer (ThermoScientific). A PCR was performed in a PTC-200 thermocycler (MJ Research) using the following steps: 94°C for 3 min and 35 cycles of 94°C for 1 min; 56°C for 30 s; 72°C for 30 s and 72°C for 5 min. For genotyping the following primers were used: *SRY* sense 5'-AGCGATGATTA-CAGTCCAGC-3' and antisense 5'-CCTACAGCTTTGTCCAGTGG-3'; *FGF16* sense 5'-CGGGAGGGA TACAGGACTAAAC-3' and antisense 5'-CTGTAGGTAGCATCTGTGGC-3'. The presence of DNA extracted from the two sexual chromosome X (*FGF16*: 495 bp) and Y (*SRY*: 538 bp) was assessed by electrophoresis on a 2% agarose gel.

The DNA was visualized thank to SybrGreen staining under an UV transilluminator (Biorad Gel Doc XR + with Image Lab Solfware) and compared against a known molecular weight marker (DNA Step Ladder 50 bp, Promega).

## Statistical analysis

Sample sizes for physiological and neuroanatomical studies and gene expression analyses were estimated based on prior experience and those represented in the extant literature. Typically, mice taken from at least two different litters for each group were used. No stringent randomization method was used to assign subjects in the experimental groups or to process data.

Quantitative RT-PCR gene expression data were analyzed using SDS 2.4.1 and Data Assist 3.0.1 software (Applied Biosystems, Carlsbad, CA). All other analyses were performed using Prism 5 (GraphPad Software). Data sets were assessed for normality (Shapiro-Wilk test) and variance. Where

appropriate a one-way or two-way ANOVA followed by post hoc testing (specified in the figure legends) was performed and for non-Gaussian distributions, a Kruskal-Wallis test followed by Dunn's multiple comparison test was used – indicated in figure legends. Exact P/adjusted p values are given in figure legends where possible. $\alpha$ was set at 0.05 for all experiments excluding WES data.

## Ethics

Animal experimentation: the study was performed in strict accordance with the Guidelines specified by the European Union Council Directive of September 22, 2010 (2010/63/EU). The protocols were approved by the Ethical Committee of the French Ministry of Education and Research (APA-FIS#13387–2017122712209790 v9).

Human fetal material: the study was approved by the French agency for biomedical research (Agence de la Biomédecine, Saint-Denis la Plaine, France, protocol n°: PFS16–002). Non-pathological human fetuses were obtained from voluntarily terminated pregnancies after obtaining written informed consent from the parents (Gynaecology Department, Jeanne de Flandre Hospital, Lille, France).

Human subjects: this study was approved by the ethics committee of the University of Lausanne, and all participants provided written informed consent prior to study participation.

## Acknowledgements

We thank M Tardivel and A Bongiovanni (microscopy core facility), M-H Gevaert (histology core facility), D Taillieu and J Devassine (animal core facility) and the BICeL core facility of the Lille University School of Medicine for expert technical assistance. This work was supported by: the Institut National de la Santé et de la Recherche Médicale (INSERM), France [grant number U1172]; Agence Nationale de la Recherche (ANR), France [ANR-14-CE12-0015-01 RoSes and GnRH to PG]; the Centre Hospitalier Régional Universitaire, CHU de Lille, France (Bonus H to PG and Ph.D. fellowship to NEHM); the European Research Council (ERC) under the European Union's Horizon 2020 research and innovation program (ERC-2016-CoG to PG grant n° 725149/REPRODAMH); Horizon 2020 Marie Sklodowska-Curie actions – European Research Fellowship (H2020-MSCA-IF-2017) to MI; the Spanish Ministerio de Ciencia e Innovación (grants CTQ2017-87889-P and CTQ2017-83745-P to LM, LAC and J-DM); the Generalitat de Catalunya (grant 2017SGR1323 to LAC and J-DM). Support of COST Action CM1306 is kindly acknowledged. LAC thanks Generalitat de Catalunya for her Ph.D. grant. LM thanks the 'Talent 2017' program from the Universitat Autònoma de Barcelona.

## Additional information

### Funding

| Funder | Grant reference number | Author |
|---|---|---|
| Agence Nationale de la Recherche | ANR-14-CE12-0015-01 | Paolo Giacobini |
| Horizon 2020 Framework Programme | ERC-2016-CoG 725149 | Paolo Giacobini |
| Ministerio de Ciencia e Innovación | CTQ2017-87889-P | Lur Alonso-Cotchico Laura Masgrau Jean-Didier Maréchal |
| Ministerio de Ciencia e Innovación | CTQ2017-83745-P | Lur Alonso-Cotchico Laura Masgrau Jean-Didier Maréchal |
| Horizon 2020 Framework Programme | H2020-MSCA-IF-2017 | Monica Imbernon |
| Generalitat de Catalunya | 2017SGR1323 | Lur Alonso-Cotchico Jean-Didier Maréchal |

The funders had no role in study design, data collection and interpretation, or the decision to submit the work for publication.

## Author contributions
Samuel Andrew Malone, Andrea Messina, Formal analysis, Investigation, Writing—original draft; Georgios E Papadakis, Resources, Data curation, Writing—original draft, Writing—review and editing; Nour El Houda Mimouni, Data curation, Investigation, Visualization; Sara Trova, Investigation, Visualization; Monica Imbernon, Pascal Pigny, Data curation, Investigation; Cecile Allet, Methodology; Irene Cimino, Lur Alonso-Cotchico, Data curation, Formal analysis, Investigation; James Acierno, Data curation, Software, Formal analysis, Writing—original draft; Daniele Cassatella, Software, Formal analysis, Validation; Cheng Xu, Investigation; Richard Quinton, Gabor Szinnai, Resources; Laura Masgrau, Formal analysis, Validation, Investigation; Jean-Didier Maréchal, Data curation, Formal analysis, Supervision; Vincent Prevot, Conceptualization, Data curation; Nelly Pitteloud, Conceptualization, Data curation, Funding acquisition, Writing—original draft, Writing—review and editing; Paolo Giacobini, Conceptualization, Formal analysis, Supervision, Funding acquisition, Validation, Writing—original draft, Project administration, Writing—review and editing

## Author ORCIDs
Samuel Andrew Malone (iD) https://orcid.org/0000-0002-6824-6854
Vincent Prevot (iD) http://orcid.org/0000-0001-7185-3615
Paolo Giacobini (iD) https://orcid.org/0000-0002-3075-1441

## Ethics
Human subjects: Human fetal material: the study was approved by the French agency for biomedical research (Agence de la Biomédecine, Saint-Denis la Plaine, France, protocol n°: PFS16-002). Non-pathological human fetuses were obtained from voluntarily terminated pregnancies after obtaining written informed consent from the parents (Gynaecology Department, Jeanne de Flandre Hospital, Lille, France). Human subjects: this study was approved by the ethics committee of the University of Lausanne, and all participants provided written informed consent prior to study participation.
Animal experimentation: Animal experimentation: the study was performed in strict accordance with the Guidelines specified by the European Union Council Directive of September 22, 2010 (2010/63/EU). The protocols were approved by the Ethical Committee of the French Ministry of Education and Research (APAFIS#13387-2017122712209790 v9).

## Decision letter and Author response
Decision letter https://doi.org/10.7554/eLife.47198.023
Author response https://doi.org/10.7554/eLife.47198.024

# Additional files

## Supplementary files
• Supplementary file 1. List of primers used for genotyping and for mutagenesis experiments.
DOI: https://doi.org/10.7554/eLife.47198.020
• Transparent reporting form
DOI: https://doi.org/10.7554/eLife.47198.021

## Data availability
All data generated or analysed during this study are included in the manuscript and supporting files. Raw data files have been provided for Figures 1, 2, 4, 5, 7, Figure 4—figure supplement 1. The human sequencing data from the patients and their family members in this study cannot be made available to prevent traceability of the patients and because not all participants gave their consent for releasing their data publicly. However, the source sequencing human data can be made available on request to the corresponding author (Dr. Nelly Pitteloud).

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
