## [Decision Letter]

Thank you for submitting your article "Defective AMH signaling disrupts GnRH neuron development and function and associates with hypogonadotropic hypogonadism" for consideration by *eLife*. Your article has been reviewed by three peer reviewers, and the evaluation has been overseen by a Reviewing Editor and Catherine Dulac as the Senior Editor. The following individuals involved in review of your submission have agreed to reveal their identity: Carol F Elias (Reviewer #1); Markey McNutt (Reviewer #2).

The reviewers have discussed the reviews with one another and the Reviewing Editor has drafted this decision to help you prepare a revised submission.

Summary:

In this study, Malone et al. reported results showing a link between defective AMH signaling and defective embryonic migration of GnRH and abnormal development of projections to basal forebrain. The study by Malone and colleagues describes the identification of loss-of-function mutation in AMH and AMHR2 in 3% of congenital HH probands. Authors showed both AMH and AMHR2 are expressed in migratory GnRH neurons of mice and humans, suggesting a role in GnRH neuronal migration during development. They also performed functional studies in mice using mouse genetics, pharmacology and ex vivo explants in developing embryos, identified relevant signaling pathways using in vitro cell lines and used modeling to infer structural changes and pathological consequences. The manuscript provides novel information that will be of wide interest. As outlined below, the biggest criticism to be dealt with is regarding the over interpretation of the clinical genetic data.

Essential revisions:

The overall data suggests a role for anti-müllerian hormone and its receptor in the migration of GnRH neurons in the mouse model shows hypogonadism when AMHR2 is disrupted. The conclusion from the paper is that the 3 variants identified in AMH and the single variant found in AMHR2 are pathogenic. However, the authors do not provide adequate explanation as to how they arrived at the classification of these variants.

Clinical variant interpretation is complex, and the widely excepted guidelines for interpretation were set forth by the American College of Medical Genetics and Genomics in PMID 25741868. Under these guidelines, all of the variants reported in this paper would be classified as variants of uncertain significance. All 4 pedigrees are small and there is not enough data to show appropriate disease segregation in any of the families.

Additionally, all of these variants have been identified in control populations and have significant frequencies relative to the known disease frequency in gnomAD. For example, the first variant in AMH (c.295A >T) has an overall population frequency of 0.02%. By the Hardy-Weinberg equation, you would expect the heterozygous carrier frequency in the population to be about 0.04%. This is well above the reported disease frequency in the paper of 1 in 4000 (0.025%). Even if this were the only variant contributing to hypogonadotrophic hypogonadism we would expect a much higher disease frequency in the population. This indicates that even if this were a pathogenic variant it would have very low penetrance. If any of these variants had a significant effect on the reproductive axis, they should be under strong negative selection and never reached such high population frequencies.

The data would be much more compelling with rare, de novo variants or in large families to establish clear segregation.

To really be compelling and provide sufficient evidence to classify these variants as likely pathogenic, second and third unrelated families with clear phenotypes and the same variants would need to be demonstrated. Based on the data from the mice and embryology, it certainly seems reasonable that AMH could be involved in Kallmann syndrome and hypogonadotrophic hypogonadism in humans. However, there is not sufficient evidence presented in the paper to reach the conclusions that they have about the pathogenicity of the reported variants. I do feel this data should be published, but it should be done so with appropriately tempered interpretation of the clinical variants.

If published with the current data, the authors should rewrite the interpretation and state the limitations as clearly as they do for the mouse work. The statements of pathogenicity should be replaced with more guarded interpretation. The in vitro data suggests that the variants observed in the clinical cases are deleterious to protein function. However, additional studies will be needed to determine if they are truly pathogenic in humans. The frequency of the identified variants in control populations suggest that if these are pathogenic variants, they are low penetrance. This would be consistent with previous observations that genes associated with KS and IHH are of reduced penetrance and may represent digenic/oligogenic inheritance.

It is suggested that the authors present the clinical data, speculate that these are candidate genes, suggest follow-up experiments and replication in other cohorts, and delete their statements of over-interpretation of variant pathogenicity. If the clinical observation is true, given the frequency of this disease and high number of cases with no identified genetic cause, it should replicate in multiple other clinical cohorts. They will still be the first to report the possible role of these genes even they do not provide definitive evidence.

---

## [Author Response]

Essential revisions:The overall data suggests a role for anti-müllerian hormone and its receptor in the migration of GnRH neurons in the mouse model shows hypogonadism when AMHR2 is disrupted. The conclusion from the paper is that the 3 variants identified in AMH and the single variant found in AMHR2 are pathogenic. However, the authors do not provide adequate explanation as to how they arrived at the classification of these variants.

We appreciate the reviewers’ perspective, and have edited the manuscript to more clearly indicate that the mutations identified are loss-of-function rather than categorizing them as pathogenic, which would imply a clinical determination.

Clinical variant interpretation is complex, and the widely excepted guidelines for interpretation were set forth by the American College of Medical Genetics and Genomics in PMID 25741868. Under these guidelines, all of the variants reported in this paper would be classified as variants of uncertain significance.

We agree that the ACMG guidelines for interpreting variants is a useful tool in the clinical setting to establish pathogenic mutations in known genes related to specific Mendelian disorders. However, we do not feel these standards are applicable to the current manuscript, based on 3 critical statements within the published ACMG guidelines [1]:

“This report recommends the use of specific standard terminology…to describe variants identified in genes that cause Mendelian disorders.”

“It is not intended for the interpretation of somatic variation, pharmacogenomic (PGx) variants, or variants in genes associated with multigenic non-Mendelian complex disorders.”

“Care must be taken when applying these rules to candidate genes (“genes of uncertain significance” (GUS)) in the context of exome or genome studies … because this guidance is not intended to fulfil the needs of the research community in its effort to identify new genes in disease.”

We thus feel that applying these clinical standards within this research study would be inadvisable, especially given the high degree of oligogenicity observed in CHH [2-4]. However, it is interesting to note that the automated classification of these variants using InterVar (wintervar.wglab.org/) was initially “Uncertain Significance”. Adding the information from our in vitro or in vivo studies (ACMG category PS3), all 4 mutations would be classified as “Likely Pathogenic”.

However, we do agree with the reviewer that additional families and larger studies in CHH patients is needed in order to confirm the contribution of AMH and AMHR2 mutations in the pathophysiology of CHH. We have updated the Discussion to reflect this.

All 4 pedigrees are small and there is not enough data to show appropriate disease segregation in any of the families.

We agree, and have added this point to the Discussion that large-scale studies needed for further confirmation.

Additionally, all of these variants have been identified in control populations and have significant frequencies relative to the known disease frequency in gnomAD. For example, the first variant in AMH (c.295A >T) has an overall population frequency of 0.02%. By the Hardy-Weinberg equation, you would expect the heterozygous carrier frequency in the population to be about 0.04%. This is well above the reported disease frequency in the paper of 1 in 4000 (0.025%). Even if this were the only variant contributing to hypogonadotrophic hypogonadism we would expect a much higher disease frequency in the population. This indicates that even if this were a pathogenic variant it would have very low penetrance. If any of these variants had a significant effect on the reproductive axis, they should be under strong negative selection and never reached such high population frequencies.

We agree with the reviewers’ statement regarding allele frequencies in controls when it is made in the context of Mendelian disorder. However, CHH is not a strictly Mendelian disorder. Oligogenicity (i.e. mutations in more than one gene) is a well-known mode of inheritance in CHH, and the increased use of next-generation sequencing has uncovered increasing degrees of oligogenicity. Indeed, very few of the >30 genes involved in CHH are highly penetrant and rare variants in these genes can found in the controls **^4^**. We have modified the Discussion to clarify this point.

*The data would be much more compelling with rare,* de novo *variants or in large families to establish clear segregation.*

We agree with the reviewers, and have noted this point in the Discussion as follows: “This is a common theme in the genetics of CHH, and factors such as digenic/oligogenic inheritance, epigenetic regulation or non-genetic contributions likely play important roles. As the current study is limited by the small number of probands harboring AMH and AMHR2 mutations, confirmation in larger CHH cohorts will be necessary to establish the specific contributions of these two genes in the pathogenesis of CHH”.

To really be compelling and provide sufficient evidence to classify these variants as likely pathogenic, second and third unrelated families with clear phenotypes and the same variants would need to be demonstrated.

As noted above, we do not feel that applying ACMG classification to these mutants is wise at this stage. However, we agree that having additional families harboring the same mutations would be optimal. But as is the case with many rare disorders, most mutations in other CHH genes are private. We have added to the Discussion to point out that large, multi-national studies are needed.

Based on the data from the mice and embryology, it certainly seems reasonable that AMH could be involved in Kallmann syndrome and hypogonadotrophic hypogonadism in humans. However, there is not sufficient evidence presented in the paper to reach the conclusions that they have about the pathogenicity of the reported variants. I do feel this data should be published, but it should be done so with appropriately tempered interpretation of the clinical variants.

We have edited the manuscript to describe the mutations as loss-of-function or damaging rather than pathogenic.

*If published with the current data, the authors should rewrite the interpretation and state the limitations as clearly as they do for the mouse work. The statements of pathogenicity should be replaced with more guarded interpretation. The* in vitro data suggests that the variants observed in the clinical cases are deleterious to protein function. However, additional studies will be needed to determine if they are truly pathogenic in humans. The frequency of the identified variants in control populations suggest that if these are pathogenic variants, they are low penetrance. This would be consistent with previous observations that genes associated with KS and IHH are of reduced penetrance and may represent digenic/oligogenic inheritance.It is suggested that the authors present the clinical data, speculate that these are candidate genes, suggest follow-up experiments and replication in other cohorts, and delete their statements of over-interpretation of variant pathogenicity. If the clinical observation is true, given the frequency of this disease and high number of cases with no identified genetic cause, it should replicate in multiple other clinical cohorts. They will still be the first to report the possible role of these genes even they do not provide definitive evidence.

We have edited the manuscript to describe the mutations as loss-of-function or damaging rather than pathogenic, and have updated the Discussion section regarding the overall interpretation of the results as related to CHH patients. In addition, we agree with the reviewer that the frequency of identified variants in controls suggests that mutations in AMH and AMHR2 are likely part of a digenic/oligogenic inheritance. We softened the final phrase in the Discussion to reflect that our work “identifies heterozygous mutations in AMH and AMHR2 in CHH patients”.

References:

1) Richards S, Aziz N, Bale S, et al. Standards and guidelines for the interpretation of sequence variants: a joint consensus recommendation of the American College of Medical Genetics and Genomics and the Association for Molecular Pathology. Genetics in medicine: official journal of the American College of Medical Genetics 2015;17:405-24.

2) Miraoui H, Dwyer AA, Sykiotis GP, et al. Mutations in FGF17, IL17RD, DUSP6, SPRY4, and FLRT3 are identified in individuals with congenital hypogonadotropic hypogonadism. Am J Hum Genet 2013;92:725-43.

3) Sykiotis GP, Plummer L, Hughes VA, et al. Oligogenic basis of isolated gonadotropin-releasing hormone deficiency. Proceedings of the National Academy of Sciences of the United States of America 2010;107:15140-4.

4) Cassatella D, Howard S, Acierno J, et al. Congenital Hypogonadotropic Hypogonadism and Constitutional Delay of Growth and Puberty Have Distinct Genetic Architectures. Eur J Endocrinol 2018.